

# Crossed product algebras and generalized entropy for subregions

**Shadi Ali Ahmad[1][*] and Ro Jefferson[2][†]**

**1** Center for Cosmology and Particle Physics, New York University, New York, NY 10003, USA
**2** Institute for Theoretical Physics, and Department of Information and Computing Sciences,
Utrecht University, Princetonplein 5, 3584 CC Utrecht, The Netherlands

[*] shadiraliahmad@gmail.com , [†] r.jefferson@uu.nl

## Abstract

An early result of algebraic quantum field theory is that the algebra of any subregion in a QFT is a von Neumann factor of type $III_1$, in which entropy cannot be well-defined because such algebras do not admit a trace or density states. However, associated to the algebra is a modular group of automorphisms characterizing the local dynamics of degrees of freedom in the region, and the crossed product of the algebra with its modular group yields a type $II_\infty$ factor, in which traces and hence von Neumann entropy can be well-defined. In this work, we generalize recent constructions of the crossed product algebra for the TFD to, in principle, arbitrary spacetime regions in arbitrary QFTs, formally paving the way to the study of entanglement entropy without UV divergences. In contrast to previous works, we emphasize that this construction is independent of gravity. In this sense, the crossed product construction represents a refinement of Haag's assignment of nets of observable algebras to spacetime regions by providing a natural construction of a type II factor. We present several concrete examples: a QFT in Rindler space, a CFT in an open ball of Minkowski space, and arbitrary boundary subregions in AdS/CFT. In the holographic setting, we provide a novel argument for why the bulk dual must be the entanglement wedge, and discuss the distinction arising from boundary modular flow between causal and entanglement wedges for excited states and disjoint regions.

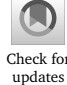

# 1   Introduction

The central object of study in algebraic quantum field theory is a so-called net of observables [1–3]

$$\mathcal{O} \to \mathcal{A}(\mathcal{O}),\tag{1}$$

where $\mathcal{O}$ is a spacetime region and $\mathcal{A}(\mathcal{O})$ is the algebra of observables localized to that region. Generally speaking, this is a C*-algebra which may be completed to a von Neumann algebra in the weak operator topology [4]. These algebras satisfy $\mathfrak{A}(\mathcal{O})'' = \mathfrak{A}(\mathcal{O})$ where the prime denotes the commutant in $\mathcal{B}(\mathcal{H})$. There is an analogous operation at the level of spacetime regions, namely the spacelike complement $\mathcal{O}'$, and regions satisfying $\mathcal{O}'' = \mathcal{O}$ are called causally complete.[1] Thus, while any open spacetime region may be assigned a C*-algebra $\mathcal{A}(\mathcal{O})$, imposing further axioms on the assignment (1) leads to relations between the von Neumann algebras defined on $\mathcal{O}$ and other related regions. In particular, if one has Haag duality, $\mathfrak{A}(\mathcal{O})' = \mathfrak{A}(\mathcal{O}')$, then $\mathfrak{A}(O) = \mathfrak{A}(\mathcal{O}'')$ for causally complete regions $O$. This will be the case for the concrete examples in the present work, though we emphasize that the abstract construction holds in generality.[2] Crucially, for any theory obeying the standard axioms of QFT, the resulting local algebra is type III$_1$ [4, 9–13]. Physically, one can think of this algebra as consisting of local observables constructed from suitably smeared fields with support limited to the region.

The type III nature of these algebras is responsible for a central difficulty in the study of entanglement in QFT, namely the infinities resulting from the fact that the von Neumann entropy is ill-defined. The infinite value of entanglement entropy has plagued the subject from the start, and is especially relevant in light of attempts to understand the entropy of black holes and horizons more generally [14]. These infinities arise from the lack of a finite trace on the algebras of observables, thereby obstructing the existence of density states localized to the region. Moreover, the algebra of operators in type III theories does not admit a tensor factorization, in contrast to type I factors appearing in quantum mechanics [15–17].[3] See [20] for a relevant review of entanglement entropy in QFT, [21] for an explanation of the type classification in terms of renormalization schemes, or [22] for a pedagogical introduction to operator algebras aimed at physicists.

Physicists have generally taken a cavalier attitude to such technicalities, and the study of entanglement in field theory has played a central role in many developments, in particular our understanding of the holographic dictionary in AdS/CFT. A key question in this context is: what bulk physics can be reconstructed from a given subregion of the boundary CFT? As we shall discuss in sec. 4, the answer to this question relies heavily on entanglement-based probes [23–25], which motivates a careful understanding of entanglement for holographic theories, particularly in the context of the emergent spacetime paradigm (cf. sec. 5). In the specific context of the TFD, which will appear in the following sections, the entanglement

---

[1]The causal completion of a region $\mathcal{O}$ is generically bigger than the domain of dependence $D(\mathcal{O})$, which will be relevant in our discussion of holography in sec. 4.

[2]Haag duality is a natural condition that one expects in any reasonable QFT, but for the specific examples considered herein, it is not an assumption, since Haag duality is known to hold for Rindler space [5,6] and CFTs [7], and hence also for the holographic dual subalgebra in the bulk; see [8] and references therein.

[3]Note however that one can still assign them weaker notions of statistical independence, cf. [18,19].

structure encoded in the fact that the two copies of the bulk theory do not factorize is essential for understanding the black hole interior, which is causally inaccessible from either boundary. In attempts to understand black holes more generally, the incorrect use of type I reasoning ultimately leads to the black hole information or firewall paradox [17, 26, 27]. It thus seems that a satisfactory theory of quantum gravity will remain beyond our reach until entanglement in QFT is properly understood.

Recently, there have been exciting advancements in this vein. In an early attempt [28] to understand black hole interiors via modular inclusions, the algebra of the TFD was simply assumed to be type III, based on the idea of subregion-subregion duality (see sec. 4), i.e., the duality with the algebra of bulk operators in the left and right exteriors. Additional support for this idea was subsequently given in [29,30], which conjectured that the change in character of the boundary algebra from type I to type III arises from the Hawking-Page transition, in which black holes become canonically favourable in the bulk; see [22, 31] for more explanation. The nature of the algebras was finally proven only recently in [32] (see also [33] for further discussion of missing information in this context). In a remarkable paper [27], it was shown that one can deform the boundary algebra perturbatively in $1/N$ from type $\mathrm{III}_1$ to type $\mathrm{II}_\infty$ by adjoining the generator of the modular automorphism group, which is dual to the ADM mass in the bulk, via the crossed product [4] (reviewed below, not to be confused with the more familiar cross product). In stark contrast to type III algebras, type II algebras *do* admit a well-defined trace, and hence a meaningful definition of von Neumann entropy for density states.[4] Indeed, the exciting result of [27] is that this allows one to obtain an expression for the generalized entropy in a system (namely, the TFD above the Hawking-Page transition) in which it was previously, in the type III theory, fundamentally ill-defined. Subsequently, this crossed product construction was applied to a microcanonical version of the TFD (which avoids the need to work perturbatively) in [34], the algebra of de Sitter space together with an observer in [35], and JT gravity in [36, 37].

In this paper, we show that this reduction from type III to type II can be generalized to in principle arbitrary subregions of arbitrary QFTs by adjoining the appropriate modular Hamiltonian. In this sense, the crossed product construction may be viewed as a refinement of (1) in the sense that it allows one to associate a trace on the local algebra of observables in any quantum field theory, i.e.,

$$\mathcal{O} \rightarrow \mathfrak{A}(\mathcal{O}) \rightarrow \widehat{\mathfrak{A}}(\mathcal{O}) \,, \tag{2}$$

where the final type II algebra $\widehat{\mathfrak{A}}(\mathcal{O})$ is obtained naturally from the type III algebra $\mathfrak{A}(\mathcal{O})$, and admits density states and a trace. The essential idea is as follows: associated to each algebra is a group of modular automorphisms generated by a modular Hamiltonian, which belongs to neither the algebra nor its commutant; physically, the modular Hamiltonian describes the dynamics local to the region [4]. Modular automorphisms of any type $\mathrm{III}_1$ factor are outer, and the crossed product of the algebra with a group of outer automorphisms yields a better-behaved type II factor [38]. Thus, incorporating the associated modular flow to a local algebra yields a well-defined trace, allowing one to define generalized entropy for subregions in any quantum field theory.[5]

The advantage of this construction can also be understood to yield a formalization of the concept of a density state for quantum field theory. Many results in QFT, particularly in the

---

[4]We note however that while von Neumann entropy in type I theories may admit an interpretation in terms of counting states, this is not the case in type II theories, so the physical interpretation of von Neumann entropy in the present situation differs; see [27] for further comment.

[5]Intuitively, the crossed product of a von Neumann algebra $\mathfrak{A}$ and a group $G$, both acting on a Hilbert space $\mathcal{H}$, is the von Neumann algebra generated by $\mathfrak{A}$ and the unitary representation of $G$. Some divergences may remain after this procedure, but these are mainly IR divergences associated with infinite horizon area and can be dealt with by using suitable regulators.

study of entanglement, are formulated in terms of the reduced density matrix of some sub-region of the theory, obtained by a partial trace over the complement and normalized in the Hilbert space trace. The issue with this in the type III$_1$ algebra is two-fold: there is no finite trace with which to normalize the density states, and there is no operator $K$ in the algebra of the subregion such that the reduced density state can be written $\rho = e^{-K}$; i.e., the reduced density matrix does not exist. The crossed product construction solves both of these issues by effectively adjoining the generator of the density states, which we call the (modular) charge to distinguish it from the full modular Hamiltonian, allowing one to define density states in the resulting type II algebra as well as providing a trace that is unique up to rescaling. We also stress that one need not assume the tensor factorization of the underlying Hilbert space into subregions and their complements to obtain such reduced states, as this is not true for the types of algebras one encounters in quantum field theory. This is in line with defining density states and entanglement adapted to algebras of observables as opposed to a particular Hilbert space representation [39–41].

The remainder of this paper is organized as follows. In sec. 2, we review the crossed product construction as formulated in [27], and highlight that the reduction to type II is an intrinsic feature of incorporating the local dynamics of the region that does not necessarily rely on any gravitational aspects as suggested in [34]. We then apply this construction to some important examples of both holographic and non-holographic QFTs. We first consider non-holographic systems in 3, and construct the type II factors for an arbitrary QFT in Rindler space (subsec. 3.1) and a CFT in an open ball of Minkowski spacetime (subsec. 3.2), and obtain the generalized entropy in both cases.[6] We then turn to holography in sec. 4, and begin by providing a novel algebraic argument for why the bulk dual of a boundary subregion must be the entanglement wedge. We consider a general boundary subregion in AdS/CFT, and apply the crossed product construction for the vacuum state of a large-$N$ gauge theory. We also discuss excited states and disjoint boundary subregions, where the entanglement and causal wedges no longer coincide and the modular flow becomes non-geometric. We speculate on some boundary diagnostics of the switchover effect in the bulk based on the appearance of a non-local mixing term in the modular charge. We close in sec. 5 with a conceptual summary and ideas for future work. Since it plays a key role in our analysis, a review of the mapping between a Rindler wedge and an open ball is provided in appendix A.

**Note added:** as this work was completed, [42] appeared with closely related ideas on generalizations of the crossed product construction mentioned above. However, there are a number of key conceptual distinctions between our respective approaches. First, as with the initial works [27,34,35], their approach relies on the concept of a normalized density state on the type III$_1$ factor in order to obtain the generalized entropy (via an expression for relative entropy that holds in theories of type I or II). In contrast, our renormalization procedure is distinct from previous approaches and immediately leads to a subtracted entropy for the type III$_1$ factors, which we show is UV finite, and hence is preferable insofar as all intermediate steps are well-defined. As we shall show, the entropy for the vacuum state of the subalgebra is obtained directly without recourse to the purely formal identities for relative entropy used previously, which can only be used to compute trivial excitations that do not alter the entangling surface; see subsec. 4.2.[7] The recovery of the vacuum area contribution is another distinction in the

---

[6]As a caveat for large-$N$ gauge theories, one must renormalize the modular Hamiltonian appropriately and work perturbatively in $N$ to apply the crossed product construction. This case will be discussed in sec. 4.

[7]That said, we do employ similar expressions as [42] in our discussion of trivially excited states in 4.2. Specifically, the right-hand sides of (54) and (55) are purely formal insofar as they involve expressions for the entropy in the original type III theory, though the left-hand sides of both expressions are UV-finite. We emphasize however that the main results do not rely on such decompositions, and that these were introduced purely to illustrate that the manifestation of such excitations can be formally understood as a difference of von Neumann entropies, and does *not* involve a new area term.

present paper, as the normalization chosen in previous works effectively removes this term of interest. A third, and perhaps the most important, commonality with previous works in [42] is the reliance on gravitational constraints, i.e., the belief that (quantum) gravity is somehow essential to transmute the type of the algebra from III to II. As we emphasize however, this is not the case: the modular automorphism for any local algebra of quantum field theory is outer, and the crossed product of such an algebra with this group is type II. In other words, the crossed product is a mathematical construction that adjoins the intrinsic dynamics, and is *a priori* independent of gravitational constraints and does not require the introduction of an auxiliary observer. (Of course, for the case of linearized quantum gravity coupled to matter, the crossed product algebra will respect the constraints; see sec. 2 for further discussion.) Finally, we discuss several examples in detail, in particular subregions defined via quantum extremal surface prescriptions in AdS/CFT. In the holographic case, we also provide a simple but novel argument that the entanglement wedge (as opposed to the causal wedge) is necessarily the bulk dual of a boundary subregion, and discuss extremal surface phase transitions in relation to the emergence of non-locality of the modular flow.

## 2 Modular Hamiltonians and the crossed product

In this section, we review the general construction introduced in [27] for defining an entropy by first reducing a type III algebra to type II via the crossed product. Intuitively, the crossed product $\rtimes$ provides a way of combining an algebra together with its dynamics (i.e., group action) into a larger algebra. The corresponding Hilbert space then provides a more natural description of the states under the group action. In the present case, we will be interested in the action of the (modular) Hamiltonian, so that in some sense the crossed product provides us with a more covariant description of the physics under (modular) time evolution. Remarkably, in the case of quantum field theories, this allows one to obtain a well-defined expression for entanglement entropy by reducing the corresponding von Neumann algebra from type III to type II [27].

Here let us briefly summarize this construction; the interested reader is encouraged to consult the original work [27] for details, or [22] for a pedagogical review. To begin, suppose $\mathfrak{A}$ acts on a Hilbert space $\mathcal{H}$, and let $T$ (not necessarily in $\mathfrak{A}$) be a self-adjoint operator that generates a group of automorphisms, i.e.,

$$e^{isT}ae^{-isT} \in \mathfrak{A}, \qquad \forall a \in \mathfrak{A}, \quad s \in \mathbb{R}. \tag{3}$$

More compactly, defining the unitary operator $U = e^{isT}$, we may express this as

$$U\mathfrak{A}U^{-1} = \mathfrak{A}. \tag{4}$$

Such automorphisms come in two types: if $U \in \mathfrak{A}$, the automorphism is called inner; otherwise, it is outer. Physically, inner automorphisms are simply unitary transformations that belong to the algebra. Accordingly, they don't result in any fundamentally new structure, in the sense that the crossed product algebra (introduced momentarily) reduces to a simple tensor product algebra. Conversely, outer automorphisms can be thought of as equivalence classes of unitary transformations. In the present case, we are specifically interested in the crossed product of the algebra and its modular automorphism group, which for type III factors is always outer. Crucially, it is a standard result in the theory of operator algebras that the crossed product of a type $III_1$ algebra with its modular automorphism group is a von Neumann algebra of type $II_\infty$ [38].

This fact was beautifully exploited in [27] to obtain a well-defined formula for the entropy

of the thermofield double (TFD) state, which is dual to the eternal black hole in AdS/CFT,[8] and subsequently applied to de Sitter space [35] and a microcanonical version of the TFD [34]; see also [36, 37] for applications to JT gravity. The general construction can be phrased as follows: let $T$ be the generator of an automorphism group of $\mathfrak{A}$ as above, and $X$ be some bounded function on $\mathbb{R}$. It is important that $T$ has a well-defined action on $\mathfrak{A}$, while $X$ belongs to the commutant (by construction). In [27, 34], $X$ was taken to be the Hamiltonian of the left CFT, $H_L$ (since one works from the right copy for convenience), and in [35] it was interpreted as the Hamiltonian of an observer in dS. In general, we will take it to be the modular charge of the commutant $\mathfrak{A}'$.[9] The crossed product algebra

$$\widehat{\mathfrak{A}} := \mathfrak{A} \rtimes \mathbb{R}, \tag{5}$$

is then obtained by adjoining (bounded functions of) $T + X$ to $\mathfrak{A}$, so that $\widehat{\mathfrak{A}}$ is generated by operators of the form [27]

$$a e^{isT} \otimes e^{isX}, \qquad a \in \mathfrak{A}, \quad s \in \mathbb{R}. \tag{6}$$

Note that if $T \in \mathfrak{A}$, then $a e^{isT} \in \mathfrak{A}$, so $\widehat{\mathfrak{A}}$ reduces $\mathfrak{A} \otimes \mathbb{R}$. In the present case, we take $T$ to be the modular Hamiltonian of the original algebra, which belongs neither to the algebra nor to the commutant.

Specifying to the case of the TFD [27], let us take $\mathfrak{A}$ to be the algebra of the right CFT.[10] We then wish to adjoin $H_R$. However, this is *not* the modular Hamiltonian, since the latter has support on both the algebra and its commutant, i.e., both CFTs. Formally, one writes

$$H = H_R - H_L, \tag{7}$$

which expresses the fact that time translations are a group of automorphisms of the TFD in which time moves upwards on the right boundary and downwards on the left. In contrast, while one can define suitably normalized one-sided Hamiltonians $H_{R,L}$ that have a well-defined action on their respective CFT, they do not generate automorphisms of the TFD, since they act only on one side. We will usually refer to these one-sided Hamiltonians as charges or generators, to avoid confusion with the modular Hamiltonian $H$. Consequently, we instead identify $T$ with (7), which generates an group of outer automorphisms of the algebra. We can then take $X := H_L$, which acts only on the commutant, so that adjoining $T + X$ is physically equivalent to adjoining $H_R$, as originally desired.[11] In the bulk of course, this corresponds to the fact that the two copies of AdS are infinitely entangled and do not form a tensor product, so the one-sided Hamiltonians are not even well-defined; see [22, 27] for further discussion.

Adjoining the one-sided Hamiltonian to the algebra in this manner was analyzed for the TFD dual to an AdS-Schwarzschild black hole in [27, 34]. In the bulk, the modular Hamiltonian is formally written $h = h_R - h_L$, and acts on both the left and right exterior algebras, corresponding to a timeshift in opposite directions, so that the evolution of the entire $t = 0$ Cauchy slice is smooth across the bifurcation surface. This is a symmetry of the spacetime that can be removed by a suitable diffeomorphism, but the latter acts non-trivially on the boundary, where the timeshift is conjugate to the ADM mass. It thus appears that in order for the algebra to

---

[8]Throughout this work, we consider the TFD above the Hawking-Page transition, where the algebra is of type III$_1$ in the thermodynamic limit.

[9]To streamline the general exposition, we are temporarily ignoring issues of normalization of these operators, which will be discussed in the concrete examples below.

[10]This was denoted $\mathfrak{A}_{r,0}$ in [27], while the crossed product algebra was denoted $\mathfrak{A}_R$. Here, we will simply use a hat to denote the crossed product algebra, and drop the subscripts.

[11]We emphasize however that since the decomposition (7) is purely formal, $T + X \neq H_R$ as an operator. Even if we manage to construct suitably normalized charges $H_L$, $H_R$, the algebra under consideration is still type III and hence does not factorize.

be invariant under these large gauge transformations, it is necessary to adjoin the ADM mass, which is dual to $h_R$ (or $h_L$). This led [34] to remark that, counterintuitively, including gravity makes entanglement entropy better defined in quantum gravity than in quantum field theory, due to the transition from type III to II above. However, one of the lessons of the present work is that gravity is not intrinsically involved in general. As mentioned previously, the crossed product is ultimately just a means of enlarging the algebra in such a way as to encode some dynamics, i.e., invariance under some group action. In AdS, one desires invariance under large gauge transformations, so it is indeed natural to adjoin gravity, i.e., the ADM mass. But in the case of Rindler space (discussed in subsec. 3.1) one wants invariance under Lorentz boosts; and in the case of an open ball in flat space, or an entanglement wedge in AdS (discussed in subsec. 3.2 and sec. 4, respectively), one wants invariance under modular time evolution. In all these cases, one adjoins the corresponding generator, but the physical interpretation differs in each, and the source of the simplification inherent in the reduction from type III to type II is not intrinsically gravitational *per se*.

As alluded above, since $\mathfrak{A}$ is a type III$_1$ von Neumann algebra, the crossed product algebra $\widehat{\mathfrak{A}}$ is type II$_\infty$ [38]. This is exciting, since von Neumann entropy is ill-defined for type III algebras, but can be defined for algebras of type II, and hence the crossed product construction is a promising means of defining entropy in quantum field theory free of the usual pathologies.[12] Indeed, it was shown in [27] how to define an entropy for states in the Hilbert space corresponding to $\widehat{\mathfrak{A}}$, which amounts to defining a trace on $\widehat{\mathfrak{A}}$. Let $\Psi \in \mathcal{H}$ be a state in the original Hilbert space corresponding to $\mathfrak{A}$, and $g(X)$ be an everywhere-positive function in $L^2(\mathbb{R})$. Then the algebra $\widehat{\mathfrak{A}}$ acts on

$$\widehat{\mathcal{H}} := \mathcal{H} \otimes L^2(\mathbb{R}). \tag{8}$$

Since we are interested in the case where $X$ corresponds to some Hamiltonian, there should be no mixing between these factors. Accordingly, we will restrict our analysis to separable states, which may be written in the form

$$\widehat{\Psi} := \Psi \otimes g(X)^{1/2} \in \widehat{\mathcal{H}}, \tag{9}$$

where the square root is chosen for later convenience.[13] Then for some $\widehat{a} \in \widehat{\mathfrak{A}}$, there exists a trace defined as [27]

$$\operatorname{Tr} \widehat{a} = \langle \widehat{\Psi} | \widehat{a} K^{-1} | \widehat{\Psi} \rangle = \int_{-\infty}^{\infty} \mathrm{d}X \, e^X \langle \Psi | \widehat{a} | \Psi \rangle, \tag{10}$$

where

$$K := e^{-(T+X)} g(T+X) \in \widehat{\mathfrak{A}}, \tag{11}$$

and the second equality of (10) as well as the form of the entropy below rely on the fact that the modular Hamiltonian annihilates the vacuum state, i.e., $T|\Psi\rangle = 0$. From the form of (10), we can then identify $K$ as the density matrix for the state $\widehat{\Psi}$. To see this, observe that replacing $\widehat{a} \mapsto \widehat{a}K$ recovers the standard formula for the expectation value of an operator $\widehat{a}$ in the state $\rho_{\widehat{\Psi}} = K$:

$$\operatorname{Tr} \rho_{\widehat{\Psi}} \widehat{a} = \langle \widehat{\Psi} | \widehat{a} | \widehat{\Psi} \rangle, \tag{12}$$

and that this satisfies the normalization condition $\operatorname{Tr} \rho_{\widehat{\Psi}} = 1$. Note however that the trace is not defined for all elements $\widehat{a}$, which is here regarded as a functional of $X$. For example, if

---

[12]Of course, the study of entanglement entropy in QFT is a rich and highly developed subject, but the technically invalid use of type I reasoning relies on *ad hoc* regularization schemes, and is ultimately responsible for the black hole information paradox [26, 27].

[13]As will become clear below, the second factor in (8) corresponds to the modular charge that we adjoin to the original algebra; the function $g$ will then gain an interpretation as the wavefunction determining quantum fluctuations in the modular charge.

$\widehat{a} \in \mathfrak{A} \subset \widehat{\mathfrak{A}}$, then the integral clearly diverges. The von Neumann entropy of the state $\widehat{\Psi}$ is then [27]

$$S(K) = -\text{Tr}\, K \ln K = \int_{-\infty}^{\infty} dX\, g(X)(X - \ln g(X)), \tag{13}$$

up to an additive, state-independent constant. The latter arises from the freedom to shift $X \mapsto X + c$ for an arbitrary $c \in \mathbb{R}$. In the case where $X$ is a Hamiltonian, this amounts to a phase change at the level of the dynamics, but is not a unitary transformation of the density matrix and hence leads to a change in the entropy. In the next sections, we will see that the von Neumann entropy of the crossed product algebra is the generalized entropy of the corresponding subregion.

For the sake of completeness, we note that there may also exist other conserved charges depending on the symmetries of the theory. Suppose these charges span a Lie algebra $\mathfrak{g}$ with associated Lie group $G$. Then, it was argued in [31] that the resultant algebra is again the crossed product of $\mathfrak{A}$ by $G$. Unlike the modular group, one expects gauge groups in quantum gravity to be compact [43–45], and the crossed product of a type III factor by a compact group is of the same type, and so does not qualitatively change the discussion. The extra charges will simply be included in the new type II algebra, allowing for contributions to the generalized entropy arising from objects like electric charge and angular momentum as consistent with the first law of black hole thermodynamics.

## 3 Generalized entropy of subregions

Here we detail the construction of the algebras, and the corresponding generalized entropies, for two basic examples: a generic QFT in Rindler space, and the domain of dependence of an open ball in flat space CFT. The former serves as a fundamental example relating entanglement entropy to properties of horizons, while the latter will be useful when studying subregions in AdS/CFT in sec. 4.

### 3.1 Rindler space

In this subsection, we consider an arbitrary type $\text{III}_1$ QFT[14] in Rindler space in $D = d + 1$ dimensions, with the metric

$$ds^2 = e^{2a\xi}(-d\eta^2 + d\xi^2) + d\Omega_{d-1}^2, \tag{14}$$

where Rindler coordinates $(\eta, \xi)$ are related to Minkowski coordinates $(t, x)$ via the following hyperbolic transformation:

$$t = \frac{1}{a}e^{a\xi}\sinh a\eta, \qquad x = \frac{1}{a}e^{a\xi}\cosh a\eta, \tag{15}$$

and $a > 0$ is a constant that parametrizes the acceleration. This is a classic example in the study of QFT in curved spacetime, which illustrates fundamental properties of horizons. In particular, taking the Minkowski representation to be in the vacuum state, it is a standard result that a Rindler observer will nonetheless detect particles with a thermal spectrum at temperature [46]

$$\beta^{-1} = \frac{a}{2\pi}. \tag{16}$$

---

[14]The remarkable result of [6], which we employ below, is that the modular flow in Rindler space is purely geometric. Consequently, our results apply for any QFT, free or interacting, in any dimension.

Note that at the horizon ($x = t$), $a \to \infty$, so the temperature appears to diverge. The thermal spectrum implies that the Rindler horizon has an associated entropy, which can be computed via the classic Euclidean method of Gibbons and Hawking [47] to yield [48]

$$S_{\mathrm{R}} = \frac{A}{4G}, \tag{17}$$

where the subscript R denotes the right Rindler wedge. In this expression, the area $A$ is an integral over the horizon, so that the total entropy is infinite, but the entropy per unit area can be well-defined.[15]

Here, we show that the methods of [27] reproduce the leading-order contribution (17) as well as the subleading contribution due to thermal fluctuations in the canonical ensemble.[16] The latter are not captured by the Euclidean approach mentioned above, since this computes the entropy relative to the flat-space background, so the fluctuations of the matter content are subtracted away. As discussed in [49], these fluctuations lead to a universal logarithmic correction that arises in all thermodynamic systems in the canonical ensemble, including black holes. To our knowledge however, they have not been explicitly addressed in the context of Rindler horizons.

Denote the algebra of observables in the right Rindler wedge by $\mathfrak{A}_R$, and that in the left by $\mathfrak{A}_L$. Clearly, $\mathfrak{A}_L = \mathfrak{A}'_R$, and by von Neumann's double commutant theorem, $\mathfrak{A}'_L = \mathfrak{A}_R$. The modular Hamiltonian $H$ was first found by Bisognano and Wichmann [6] to be related the Lorentz boost generator $L$, which one can express as an integral of the stress tensor over the $t = 0$ Cauchy slice:

$$L = \int_{t=0} \mathrm{d}x \, x \, T_{00}. \tag{18}$$

The modular Hamiltonian for Rindler space is simply related to the boost generator by

$$H = 2\pi L, \tag{19}$$

where the factor of $\beta = 2\pi$ is the inverse temperature of the Unruh effect; see for example [50] for further exposition.[17] Note that $H$ acts on both the algebra $\mathfrak{A}_R$ and the commutant $\mathfrak{A}_L$; formally, one often writes

$$H = H_R - H_L, \tag{20}$$

where $H_{R,L}$ correspond to the one-sided Lorentz boosts. Intuitively, the modular Hamiltonian (18) evolves the $t = 0$ Cauchy slice upwards on the right and downwards on the left, but evolving with only $H_R$ or $H_L$ would result in a kink at the origin, so the latter do not generate an automorphism of the algebra. The fact that the one-sided charges $H_R, H_L$ generate singular states reflects the fact that evolution with only one of these operators in the type III algebra is pathological, insofar as they cannot be used to generate well-defined states in the corresponding algebras $\mathfrak{A}_R, \mathfrak{A}_L$. In this sense, the point of the crossed product construction is that suitably normalized versions of these one-sided charges do belong to the type II algebras, and can be used to describe dynamics. One way to obtain well-defined operators is by simply subtracting the average energy,

$$H_R - \langle H_R \rangle, \tag{21}$$

---

[15]The integral is performed over the codimension-2 surface parametrized by $\mathrm{d}\Omega^2_{d-1}$. In $1+1$ dimensions, this is just a point, and the area vanishes; in this case the logarithmic corrections discussed below are the leading contribution to the entropy.

[16]We work semi-classically, so that the canonical ensemble in asymptotically flat space is well-defined. This is not true for gravitational systems in asymptotically flat space, since black holes have negative specific heat.

[17]Note that some authors, including Witten and collaborators, absorb the factor of $2\pi$ into the definition of $H$; in particular, the modular Hamiltonian $\hat{h}$ in [34] is in fact the Lorentz boost $L$ in our notation.

and similarly for $H_L$; see [27, 34] for further discussion of this issue. Note that the expectation value in this expression is purely formal, as it represents the energy in the type III theory, where the trace is not defined.

This subtraction scheme for the operator normalization was used in [27, 34] and related works.[18] While it provides a clean one-sided charge, it has the severe disadvantage of eliminating the vacuum contribution in the eventual generalized entropy formula. This is problematic since, as discussed in more detail in subsec. 4.2, this is the leading area contribution defined self-consistently by the definition of the subregion for the corresponding state.[19] As a result, previous works (with the exception of [42], who utilize covariant phase space methods, cf. the discussion below (26)) resort to a slightly dubious argument adapted from Wall [51] involving the notion of von Neumann entropy (in the type III theory) to recover an area term for an "excited state." Concretely, this scheme would yield zero for the area contribution from the Rindler horizon, in contradiction to the known result above.

However, recall from sec. 2 that there is an inherent ambiguity in the entropy due to the freedom to shift $X \rightarrow X + c$. In particular, this shift may be infinite, and indeed in [34] it diverges as $G \rightarrow 0$. Since any $c$-number is in the commutant of $\mathfrak{A}$ by definition, we are free to absorb the vev into this constant, and instead work directly with $H_{R,L}$. The essential point is that one should not use the subtraction scheme (21), which has zero expectation value. As we shall see, this allows us to obtain the correct area contribution in the vacuum state for the subregion. The contribution from the infinite energy of the vacuum then appears as a state-independent constant in the entropy, which one ignores.

For concreteness, let us work from the perspective of the right Rindler wedge. Since the algebra is type III$_1$, the density matrix $\rho_R$ does not exist. As discussed in sec. 2 however, we can enlarge our algebra by adjoining the modular Hamiltonian (18) in such a way as to obtain a type II$_\infty$ algebra that does split into a left and right factor. Hence, taking $T = H$ and $X = H_L$, we obtain the crossed product algebra

$$\widehat{\mathfrak{A}}_R = \mathfrak{A}_R \rtimes \mathbb{R}, \tag{22}$$

whose corresponding Hilbert space is of the form (8). Since $\widehat{\mathfrak{A}}_R$ is a type II$_\infty$ factor, the corresponding density matrix exists, but takes the form

$$\widehat{\rho}_R = e^{-H_R} g(H_R), \tag{23}$$

cf. (11), with $T = H_R - H_L$ and $X = H_L$.[20] As a consistency check, note that this satisfies the normalization condition $\text{Tr}\,\widehat{\rho}_R = 1$; this is not apparent from the r.h.s. of (10), but can be easily verified from the intermediate form

$$\text{Tr}\,\widehat{\rho}_R = \int_{-\infty}^{\infty} dH_L \, g(H_L) = 1, \tag{24}$$

where we used the fact that $\langle \Psi | \Psi \rangle = 1$ and $g(H_L)^{1/2} \in L^2(\mathbb{R})$ by construction. The entropy

---

[18]In the case where the algebra is a large-$N$ gauge theory, additional care is required in the canonical ensemble; this will be discussed in sec. 4.

[19]That is, the modular charge knows about all relevant excitations in the subregion by definition, since these would affect the location of the entangling surface which defines the subalgebra under consideration. The corresponding area contribution below is thus defined by the cyclic and separating state that sets the location of the surface; this may be the global vacuum, or some excited state. Crucially however, one does not need to consider relative entropy to obtain the relevant area contribution; rather, this only gives the entropy difference for trivial excitations, which is why we put "excited states" in quotes in this paragraph. See subsec. 4.2 for further explanation.

[20]We emphasize again that $T + X \neq H_L$, since $T$ is really the modular Hamiltonian, cf. footnote 11. However, the formal decomposition is convenient when considering its action on the state $|\widehat{\Psi}\rangle$.

(13) is then

$$
\begin{aligned}
S(\widehat{\rho}_R) &= -\mathrm{Tr}\,\widehat{\rho}_R \ln \widehat{\rho}_R \\
&= -\int_{-\infty}^{\infty} \mathrm{d}H_L \langle \Psi | g(H_L) \ln \widehat{\rho}_R | \Psi \rangle \\
&= \int_{-\infty}^{\infty} \mathrm{d}H_L \langle \Psi | g(H_L) [H_R - \ln g(H_R)] | \Psi \rangle \\
&= \langle \widehat{\Psi} | H_R | \widehat{\Psi} \rangle - \langle \widehat{\Psi} | \ln g(H_R) | \widehat{\Psi} \rangle .
\end{aligned}
\tag{25}
$$

The final step is to relate the expectation value of $H_R$ to the horizon area. There are a few different ways of seeing this. In the proof of the generalized second law by Wall [51], a semiclassical argument is used to argue that the area of a horizon is proportional to the boost generator thereon,

$$
A = 8\pi G \langle K \rangle .
\tag{26}
$$

The argument for this is reviewed in [34], where (25) was computed for a microcanonical version of the eternal AdS black hole. Slightly more generally, this can be derived using covariant phase space formalism. In particular, [52] shows that the generator of Euclidean rotations about an entangling surface (e.g., the Rindler horizon considered here, or the extremal surfaces considered in sec. 4) is given by a quarter of the area in Planck units. Further support for this connection comes from a topological argument in the case of Euclidean black holes [53, 54], which relates the Einstein-Hilbert action to the Gauss-Bonnet theorem. In brief, the idea is that the deficit angle around the horizon, treated as a cusp in $\mathbb{R}^2 \times S^{d-1}$, is canonically conjugate to the area of the $S^{d-1}$ at that point. But this is simply a rotation in Euclidean time, which is generated by the boost operator $K$ above. Intuitively, the relation (26) reflects the fact that the generator of the modular flow encodes properties of the horizon, and in particular the entanglement structure between $\mathfrak{A}$ and $\mathfrak{A}'$. This points to a further role played by the Euclidean path integral, namely it implements the relation between the modular Hamiltonian and the area of the entangling surface semiclassically.

In our notation, $K = L_R$ in (26), thus (25) is

$$
S(\widehat{\rho}_R) = \frac{A}{4G} - \langle \widehat{\Psi} | \ln g(H_R) | \widehat{\Psi} \rangle .
\tag{27}
$$

The first term is the usual contribution (17) to the entropy from the horizon area, while the second term is due to thermal fluctuations in the canonical ensemble, which physically correspond to fluctuations in the area of the surface. This second term was already recognized by [27] as the universal contribution that arises in any thermodynamic system in the canonical ensemble [49]. Here, it can be understood as the entropy of the type I algebra of the modular Hamiltonian we adjoined above. That is, formally, one writes[21]

$$
\widehat{\rho} = e^{-\widehat{K}} ,
\tag{28}
$$

for some density matrix $\widehat{\rho}$ and generator $\widehat{K}$. For (23), this decomposes into $\widehat{K} = H_R - \ln g(H_R)$, corresponding to the two terms in (27). In this sense, $-\ln g(H_R)$ is the generator of the modular flow on $L^2(\mathbb{R})$, so the subleading contribution to the entropy arises from fluctuations in the energy of the state.

Of course, in the case of a Rindler horizon, as well as the entangling surfaces we will consider in sec. 4, the area is infinite and requires some renormalization, but this is simply the IR divergence associated to the infinite extent of the surface, and is well-understood. In flat

---

[21]In the physics literature, it is common for $\widehat{K}$ to be referred to as the "modular Hamiltonian", though this is technically incorrect when $\rho$ is the reduced density matrix on some subregion. For this reason, we have taken care to refer to this as the (one-sided) charge or generator, to distinguish from the true modular Hamiltonian $\widehat{K} - \widehat{K}'$.

space for example, one typically considers energy densities, while in AdS this is accomplished with the use of a regulator.

Note that this is formally the same result obtained in [34] and related works. However, there is a key conceptual difference stemming from our interpretation of the modular operator, namely that previous works consider the entropy of an excited state, and normalize the modular charge so as to remove the vacuum contribution. As argued above and in subsec. 4.2 below, this is slightly strange, since any excitation sufficient to affect the location of the entangling surface is self-consistently included in the modular charge by construction. It is however relevant if one wishes to consider extremely light insertions that do not affect the entangling surface (which we call trivial excitations), and we comment on this further in the context of holography in subsec 4.2. In this case on merely obtains an extra contribution due to the entropy difference in the type III theory. Notwithstanding this slight but important conceptual shift, our primary goal is simply to demonstrate that the essential methods developed in [27,34,35] are more general than previously believed, as the relevant operator to adjoin to the original type III subalgebra is the modular Hamiltonian. Hence, as in these previous works, the advancement in the present case is a direct relation of the generalized entropy, including both the leading Bekenstein-Hawking area term and the subleading thermal fluctuations, to the underlying nature of the (type II) von Neumann subalgebra. Specifically, we have thus far considered the algebra of observables in the right Rindler wedge. In the next subsection, we use the result of Casini, Huerta, and Myers [55] to transport this analysis to the domain of dependence of an open ball in a CFT.

## 3.2 Domain of dependence of an open ball

In this subsection, we consider an open ball in Minkowski space, denoted $B^{d-1}$, whose entangling surface is the codimension 2 sphere $S^{d-2}$. One can associate suitably smeared operators to such a region to form a von Neumann algebra, which might *a priori* fail to capture all operators in causal contact with the ball. However, as mentioned in the introduction, Haag duality is known to hold for CFTs, which provides an isomorphism with the algebra on the ball's causal completion, which in this case is equal to the causal domain of dependence $D := (B^{d-1})''$. As above, we denote the algebra of observables localized to this subregion as $\mathfrak{A}$, and – for any theory obeying standard AQFT axioms – it is type $\text{III}_1$ [9].

Since the construction of the crossed product algebra relies on the modular Hamiltonian, we restrict our attention to cases for which the modular Hamiltonian is known. In the case of a CFT, there exists a special conformal transformation that maps the right Rindler wedge to $D$ [55–57], which can be used to obtain the generator of the modular flow for the latter. Following the notation in [55], the conformal transformation accomplishing this is

$$x^\mu = \frac{X^\mu - X^2 C^\mu}{1 - 2X \cdot C + X^2 C^2} + 2R^2 C^\mu, \tag{29}$$

where $C^\mu = (0, -\frac{1}{2R}, 0, 0)$, $X^\mu$ are Rindler coordinates, and $x^\mu$ are coordinates on $D$.[22] Note that as a consequence of this mapping, the type III (II) algebra associated to $D$ is isomorphic to the type III (II) algebra for the right Rindler wedge above. As reviewed in appendix A, the generator of the modular flow for the region $D$ is then obtained by considering the transformation of the modular flow in the right Rindler wedge under (29); the result is [55]

$$H_D = \pi \int_B d^{d-1}x \, \frac{R^2 - r^2}{R} T^{00}(x) + c, \tag{30}$$

---

[22]Note that there is a sign error in [55]; see appendix A.

where $B$ denotes an integral over the ball $B^{d-1}$, $T^{00}(x)$ is the traceless energy-momentum tensor of the CFT, and $c$ is a constant ensuring normalization of the trace. As reviewed above, the entropy for the type II crossed product algebra will only be defined up to an additive constant, so we may drop this henceforth.

An important fact that is usually overlooked in the physics literature is that the modular Hamiltonian $H$ includes contributions from both the local subregion of interest *and* its complement. Indeed, this is the $T$ appearing in the crossed product construction reviewed in sec. 2 Thus, we also need the generator of the modular flow for $D'$, the domain of dependence of all points spacelike separated from $B^{d-1}$. Fortunately, the complement of the right Rindler wedge is the left Rindler wedge, whose Lorentz boost is the negation of the Lorentz boost for the right region. Thus, upon applying the conformal transformation in (29), one finds

$$H_{D'} = -\pi \int_{\bar{B}} d^{d-1}x \frac{R^2 - r^2}{R} T^{00}(x) + C,$$
(31)

where $\bar{B}$ is the complement of the ball. Note that in $1+1$ dimensions, this will be a disjoint interval $r = x^1 \in (-\infty, -R) \cup (R, \infty)$; the mapping of the left Rindler wedge to two disjoint intervals occurs under (29) due to the discontinuity at $-2R$ (see appendix A). Thus, the modular Hamiltonian for any CFT in the causal domain of dependence of an open ball is[23]

$$H = H_D - H_{D'}.$$
(32)

We emphasize that since the Rindler flow is purely geometric, the flow generated by (32) holds for an arbitrary CFT in the vacuum state. Intuitively, we would expect the flow in $D'$ to look approximately Rindler near the causal horizons. Expanding the integrand of $H_{D'}$ near $r = \pm R$, we have

$$-\pi \frac{R^2 - r^2}{R} = -2\pi(R \mp r) + \mathcal{O}((r \mp R)^2).$$
(33)

Hence, to leading order, $H_{D'}$ indeed takes the form of a Lorentz boost, with a domain of integration shifted by $\pm R$.

The crossed product construction then proceeds precisely as above: upon adjoining $T = H$ to the type III$_1$ algebra $\mathfrak{A}$, we obtain a type II$_\infty$ algebra $\widehat{\mathfrak{A}} = \mathfrak{A} \rtimes \mathbb{R}$, whose generalized entropy is

$$\begin{aligned}
S(\widehat{\rho}_D) &= \langle \widehat{\Psi} | H_D | \widehat{\Psi} \rangle - \langle \widehat{\Psi} | \ln g(H_D) | \widehat{\Psi} \rangle \\
&= \frac{A(\partial D)}{4G} - \langle \widehat{\Psi} | \ln g(H_D) | \widehat{\Psi} \rangle,
\end{aligned}$$
(34)

where $\widehat{\rho}_D$ is the reduced density matrix for the crossed product subalgebra $\mathfrak{A}$ of the region $D$ in the vacuum state, and $\partial D$ is the spatial boundary of the ball $B^{d-1}$. As mentioned in the previous subsection, adding matter would again yield an additional contribution from the relative entropy between the excited and vacuum states [34].

As before, the leading term is the area contribution from the entangling surface, while the subleading logarithmic term arises from thermal fluctuations. Before moving on to the next section however, it is interesting to compare this to the case of a $1+1$ dimensional CFT, where the entanglement entropy of an interval in the vacuum has a simple expression [58,59]

$$S = \frac{c}{3} \log \frac{\ell}{a},$$
(35)

where $\ell$ is the length of the interval, $a$ is an ultraviolet cutoff, and $c$ is the central charge; see [60] for a pedagogical review. Since the entangling surface in this case is just a pair of points,

---

[23]To avoid a proliferation of subscripts, we use $H$ to refer to the modular Hamiltonian in any system, which should be clear from context. Obviously, $H$ in (32) is not the same as $H$ in (20).

the area contribution to the generalized entropy (34) vanishes, so the entropy (35) must be contained in the universal logarithmic term.[24] However, a direct comparison is complicated by the fact that the entropies are not computed for states in the same algebra, since (34) applies to the type $II_\infty$ crossed product algebra obtained by adjoining the modular Hamiltonian, while (35) should apply to the original type $III_1$ algebra in the presence of a cutoff.[25] Additionally, as discussed in the previous subsection, the logarithmic term in the crossed product construction can be thought of as the generator associated to the type I algebra $L^2(\mathbb{R})$. The fluctuations in the horizon area could physically be caused by fluctuations in the bulk matter fields, but we have not done a calculation that would support this. It would be interesting to make this connection explicit, but we leave this for future work.

We note that in recent work [42], compact subregions were associated to type $II_1$ factors, whereas the algebra of the compact ball considered here is type $II_\infty$. The reason for the discrepancy is that [42] impose a positivity condition on the auxiliary observer they adjoin to the original type $III_1$ factor. In contrast, we have adjoined the modular charge, which is a field-theoretic operator. Negative energy for such operators is a subtle issue, and has led to renewed interest in quantum energy inequalities; see, e.g., [61,62] for reviews. Said differently, while it is reasonable to impose positivity on a quantum mechanical observer, it would be unnatural to project out half the modular group, which as we have emphasized is the canonical outer automorphism on the algebra.

To summarize, we use the well-known fact that the Rindler wedge in $d + 1$ dimensions is conformally related to the domain of dependence of a ball $B^{d-1}$ to construct a type II subalgebra of observables $\widehat{\mathfrak{A}}$ for any CFT in the vacuum by adjoining the modular Hamiltonian (32). The generalized entropy for states of the form (9) can then be computed using the method introduced in [27]. In addition to further generalizing the crossed product construction, this particular example plays a central role in AdS/CFT as the domain of dependence of a subregion of the boundary theory, which is dual to the entanglement wedge in the bulk. This is the application to which we turn in the next section.

## 4 Subregions in AdS/CFT

Given the importance of entanglement-based probes in AdS/CFT, it is of fundamental interest to understand the entropy associated to certain localized subregions in holography. In particular, given a spacelike interval on the boundary CFT, whose domain of dependence is the region $D$ discussed in subsec. 3.2, the bulk dual of $D$ is the entanglement wedge, defined as the causal completion of the region bounded by the bulk extremal surface anchored at the boundary of the interval. A core tenet of AdS/CFT, called subregion-subregion duality, is that the physics contained within the entanglement wedge can be reconstructed within the boundary causal diamond $D$. In algebraic language, this implies that the local subalgebra in the entanglement wedge must be dual to the boundary algebra of $D$.[26]

For the purposes of understanding entanglement entropy however, this is not sufficient, as the local algebra of the entanglement wedge is still of type III. Thus, in this subsection, we discuss the application of the crossed product construction to quantum field theories localized to subregions in AdS. The distinction between this exposition and the previous section is that one

---

[24]This term is universal insofar as it does not depend on the CFT state, only on the state $g^{\frac{1}{2}}(H_D) \in L^2(\mathbb{R})$ of the new Hilbert space upon which the crossed product algebra acts. In this sense, it is universal for the original type $III_1$ subalgebra, but not for the type $II_\infty$ subalgebra.

[25]We say "should apply" because strictly speaking, the density matrix is not well-defined for type III algebras.

[26]This was recently popularized under the name "subalgebra-subregion duality" in [63], though the duality of the bulk and boundary von Neumann algebras was previously stated in this context in [28], and was implicitly assumed in earlier work on modular flow [64,65].

can analyze the construction from both the boundary and bulk perspectives due to holography, and in particular, we assume we have a conformal field theory with an $SU(N)$ gauge group localized to the boundary of AdS. We are interested in algebras of observables $\mathfrak{A}_B$ restricted to a boundary subregion $D$, and the algebra of observables $\mathfrak{A}_b$ localized to the dual region in the bulk.

Before proceeding with the application of the crossed product construction, let us discuss the dual region in more detail. A question that immediately arises, and which generated significant activity in the years following the seminal work of Ryu and Takayanagi [23] (see for example [24, 25, 66–70]), is whether the appropriate bulk dual of $D$ is the entanglement wedge or the causal wedge. While this question has been settled in favour of the former, it is illuminating to cast this in algebraic language, which provides a more fundamental explanation for the utility of quantum error correcting codes in bulk reconstruction [28, 63].

In the boundary, the type III$_1$ algebra of observables $\mathfrak{A}_B$ localized to some subregion $B$ (taken to be $D$ above) consists of suitably smeared single-trace operators.[27] This is dual to the algebra $\mathfrak{A}_b$ of suitably smeared bulk operators in some region $b$.[28] There are *a priori* two obvious candidates for $b$: the causal wedge, defined by the bulk domain of dependence of null rays fired into the bulk from the boundary causal diamond, or the entanglement wedge, defined as the causal completion of the codimension 1 bulk subregion bounded by the extremal surface and homologous to the boundary subregion. For a single region $B$ in the vacuum state, the causal and entanglement wedges coincide. In simple cases, bulk operators in $b$ can then be represented in terms of an integral over the boundary region via the HKLL prescription [71, 72]. However, for excited states, or disjoint intervals, the causal and entanglement wedges are generically different: the entanglement wedge will generically have access to more of the AdS spacetime, which implies that the operator content may differ in the two regions. The question is then which of these is dual to the region $B$ on the boundary, i.e., what is the bulk dual of $\mathfrak{A}_B$?

In the context of bulk reconstruction, one typically considers causally complete regions in the boundary—for example, the causal diamond of a given spacelike interval. Since, as mentioned previously, Haag duality holds for any boundary subalgebra $\mathfrak{A}_B$ in the CFT, this implies that $\mathfrak{A}_B = \mathfrak{A}_{B''}$. The duality between bulk and boundary subalgebras then suggests that the corresponding bulk subalgebra should satisfy this as well, i.e., $\mathfrak{A}_b = \mathfrak{A}_{b''}$. This is trivially satisfied if $b$ is also causally complete, i.e., $b = b''$. While this is technically a sufficient rather than necessary condition, it is the most natural from the holographic perspective above.[29] The causal wedge, however, is *not* causally complete. This is a consequence of the fact that the conformal boundary is timelike rather than spacelike, which implies that there are technically no Cauchy surfaces in AdS since null rays can always propagate around any spacelike surface at infinity [73]. Thus, if we demand the same relations between algebras/subregions and their commutants on both sides of the duality, then the type III$_1$ subalgebra $\mathfrak{A}_b$ dual to the boundary algebra $\mathfrak{A}_B$ cannot be the algebra of the causal wedge.

Of course, one could argue that this technicality has an equally technical fix: simply take the causal completion of the bulk causal wedge. And in fact, in vacuum, this is precisely the entanglement wedge. To see why the causal completion of the causal wedge cannot be the bulk dual in general however, consider the case where the boundary region $B$ consists of (the causal domain of dependence of) two disjoint intervals in the vacuum state. Provided the total

---

[27]Note that as we are considering subregions of the full spacetime, the associated algebra is type III at $N \to \infty$ regardless of whether we are above or below the Hawking-Page transition.

[28]For concreteness, we are considering bulk regions $b$ that do not penetrate the horizons of any AdS black holes, in which case $\mathfrak{A}_B$ consists of light CFT operators dual to exterior fields in the bulk algebra $\mathfrak{A}_b$ [28].

[29]That is, given Haag duality, $b = b''$ immediately implies $\mathfrak{A}_b = \mathfrak{A}_{b''}$, but the converse does not necessarily hold. By "natural", we mean that this ensures the same relations between algebras resp. subregions and their commutants resp. complements on both sides of the holographic dictionary.

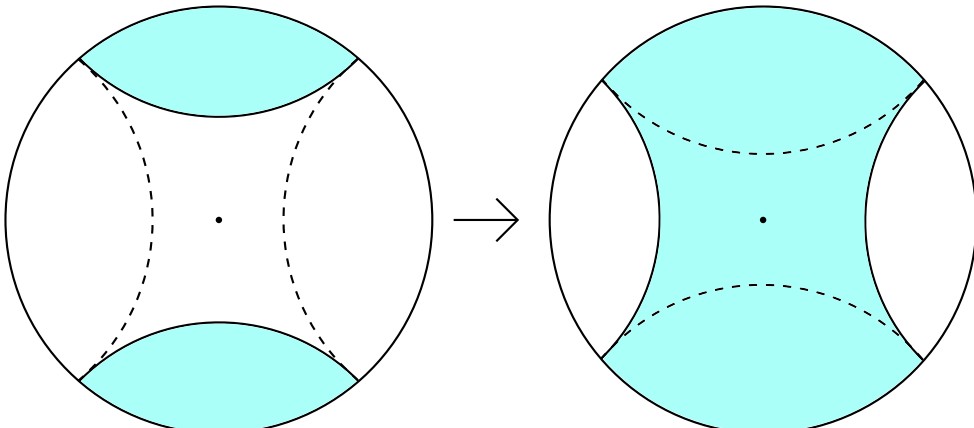

Figure 1: The bulk algebra $\mathfrak{A}_b$ is shown in blue, while the commutant $\mathfrak{A}_b' = \mathfrak{A}_{b'}$ is shown in white The dotted lines show two possible sets of entangling surfaces, which exchange dominance when their areas become equal, leading to the switchover effect. Figure taken from [74].

spacelike extent of these two intervals is less than half the boundary, the bulk dual $b$ will consist of two disjoint regions as shown in the left panel of fig. 1. When the spacelike extent exceeds half the boundary however, there is a discontinuous change in the bulk due to a switchover in which of the two sets of possible entangling surfaces becomes minimal [74]. The bulk dual $b$ after the switchover is illustrated in the right panel of fig. 1. Note that the causal wedge does not undergo a switchover, since it is not defined by any extremality condition. In this case, the question of whether to use the causal or entanglement wedge has profound consequences, since only in the latter case can we reconstruct the operator in the center of the bulk (indicated by the black dot) in $\mathfrak{A}_B$.

The key observation is that modular theory gives a natural isomorphism between $\mathfrak{A}_B$ and $\mathfrak{A}_B'$. Furthermore, by Haag duality, $\mathfrak{A}_B' = \mathfrak{A}_{B'}$. The same isomorphism holds for the bulk algebras $\mathfrak{A}_b$ and $\mathfrak{A}_b' = \mathfrak{A}_{b'}$. Therefore, the duality between the bulk and boundary algebras must be symmetric under the exchange of $B$ ($b$) and $B'$ ($b'$). But as illustrated in fig. 1, if $b$ consists of the two blue wedges, then the complement $b'$ is the single connected region in white, which implies that after the switchover, $b$ must be the single connected region in blue. Thus, in order for subregion-subregion duality to respect the isomorphism of the associated von Neumann algebras with their commutants, the bulk dual $b$ must undergo the switchover effect, which rules out the (causal completion of the) causal wedge. Simply put, the bulk dual of the boundary subregion must obey Haag duality, and the causal wedge does not.

While the above suffices to rule out the causal wedge as the bulk dual of $B$ on purely algebraic grounds, we have not addressed the possibility for some third option distinct from both the causal and entanglement wedges. This is due to the fact that the area of a non-extremal surface[30] is not gauge-invariant [65]. That is, the action of diffeomorphisms along the entangling surface must trivialize, which happens only if the surface is extremal [75]. This implies that $b$ must be the entanglement wedge. Note that if we consider the full boundary as in [34], the modular group is simply time translations of the CFT, which are diffeomorphisms in the bulk.

In the remainder of this section, we consider three important cases: a single region in

---

[30]When the causal and entanglement wedges differ, the causal surface is not extremal. It would be interesting to understand the implications of this from the algebraic perspective; e.g., what is the bulk subregion obtainable from acting with the entanglement wedge modular flow on only the causal wedge, and how does it compare to the causal completion of the latter?

vacuum AdS (described by AdS-Rindler), an excited state, and disjoint boundary subregions. In the case of AdS-Rindler, the entanglement and (causal completion of the) causal wedges coincide, reflecting the fact that the modular flow is geometric. As mentioned above however, for an excited state or disjoint regions, the distinction is essential, as the entanglement wedge is larger and, in the case of disjoint regions, exhibits the switchover effect.

## 4.1 AdS-Rindler

Consider the causal diamond $B = D$ of a spherical subregion in the boundary CFT, whose dual bulk subregion $b$ is the entanglement wedge anchored to the boundary of the sphere. In the vacuum state, $b$ is equivalent to the causal completion of the causal wedge formed by sending null rays into the bulk from the boundary domain of dependence $B$. While the size of the bulk wedge will of course depend on the size of the boundary interval, any causal wedge is related to an AdS-Rindler wedge by the isometries of AdS. Hence, without loss of generality, we may consider the spacelike extent of $B$ to be exactly half the boundary, so that $b$ is half the AdS-Rindler spacetime; see for example [76,77]. The metric for the latter may be written [78]

$$\mathrm{d}s^2 = -(\rho^2 - 1)\mathrm{d}\tau^2 + \frac{\mathrm{d}\rho^2}{\rho^2 - 1} + \rho^2 \mathrm{d}H_{d-2}^2, \tag{36}$$

where $\rho > 1$, $\tau \in \mathbb{R}$, and $\mathrm{d}H_{d-2}^2$ is the metric on the hyperbolic ball. The conformal transformations on the boundary region are in one-to-one correspondence with isometries of AdS, and the associated conserved charges may be expressed in terms of an integral over the boundary stress tensor as [27,79]

$$H = \int T_{\mu\nu}\chi^\mu \mathrm{d}V^\nu, \tag{37}$$

where $\chi^\mu$ is the boundary conformal Killing vector, and $\mathrm{d}V^\nu$ is an infinitesimal volume element of the boundary CFT. While we have written this in terms of CFT quantities, it is also the generator of the corresponding isometry in the bulk, which will be denoted $h$. We emphasize however that as an expression for the modular Hamiltonian, (37) holds only for a single connected region in vacuum, where the flow is geometric.[31] As discussed above, for excited states, or disjoint regions whose total area is more than half the boundary, the bulk modular Hamiltonian cannot be written as a simple AdS-Rindler boost, since this generates flows in the causal rather than entanglement wedge. In this subsection however, the two coincide, so $H$ takes the form of (37).

Formally, one would like to decompose (37) over the region $B$ ($b$) and its complement $B'$ ($b'$) as

$$H = H_B - H_{B'}, \tag{38}$$

cf. (20) and (32). However, in the present case of a large-$N$ gauge theory, the one-sided charges $H_B$ and $H_{B'}$ are not necessarily well-defined. This was recently explained in [27] in the context of the TFD (see also [31]), in which we have two copies of the CFT denoted right ($R$) and left ($L$). As reviewed in sec. 2, in that case $B = R$ and $B' = L$, and $H$ is the generator of time translation on both sides. Since $H$ annihilates the vacuum state, it does not belong to either $\mathfrak{A}_R$ or $\mathfrak{A}_L$, but generates an outer automorphism of both algebras. Above the Hawking-Page transition however, the one-sided charges $H_R$ and $H_L$ are not well-defined because they exhibit fluctuations that scale like $N^2$,

$$\langle H_B^2 \rangle_c = \langle H_B^2 \rangle - \langle H_B \rangle^2 \sim N^2, \tag{39}$$

---

[31]In the absence of gravitational dressing, this was proven in [80]. Since dressing ensures diffeomorphism invariance in the bulk, the algebraic argument for the entanglement wedge above still applies perturbatively in $1/N$; see below. The Killing field generating the geometric flow in the bulk reduces to $\chi^\mu$ on the boundary, hence the formal equivalence of $H$ and $h$.

where $\langle \ldots \rangle_c$ denotes the connected correlator; similarly for $B'$. This problematic scaling is connected with the type III algebra of the boundary CFT that emerges at the Hawking-Page transition, when the type I theory at lower temperatures undergoes a deconfining phase transition [22, 29]. The bulk reflection of this fact is that the TFD is dual to the eternal AdS black hole, which does not factorize into a right and left copy due to the infinite entanglement across the horizon, and consequently the corresponding bulk charges $h_R$, $h_L$ are not well-defined as operators. Said differently, evolving with these Hamiltonians would generate singular states at the bifurcation surface.

In the present work, we take $B$ to be a subregion of the boundary CFT. Conceptually, since local subalgebras in quantum field theory are always of type III$_1$, our analysis applies regardless of whether this CFT is a factor of the TFD[32] or vacuum CFT dual to empty AdS. Concretely however, we will work in vacuum where the AdS-Rindler decomposition above may be simply written down. The modular Hamiltonian $H$, which acts on both the algebra $\mathfrak{A}_B$ and its commutant $\mathfrak{A}_{B'}$, is well-defined since it annihilates the vacuum state. At large-$N$ however, the one-sided charges appearing on the right-hand side of (38) will again exhibit fluctuations that scale like $N^2$, and hence are not well-defined. Note that the fluctuations at large-$N$ are distinct from the fluctuations in the canonical ensemble responsible for the universal logarithmic term discussed above. The latter may be understood from the fact that the local charges $H_B$, $H_{B'}$ do not annihilate the local vacuum state within their respective regions, and hence may exhibit fluctuations corresponding to fluctuations of the location of the horizon, and hence of the area thereof; see [79] for a recent analysis. In contrast, the fluctuations (39) are analogous to the fact that in quantum statistical mechanics at finite temperature, thermal fluctuations diverge as the volume of the region goes to infinity. Since the volume of the entanglement wedge for any subregion in AdS is still infinite, the local charge $H_B$ therein will exhibit these pathological fluctuations even for CFTs in the vacuum state.

Note that this was not an issue in sec. 3, since we were ultimately considering subregions in the Minkowski vacuum at zero temperature, and our CFT was not taken to be a large-$N$ gauge theory. While the Rindler decomposition is superficially similar to the TFD, the temperature is proportional to the acceleration of a Rindler observer and vanishes at spatial infinity. Therefore, there are no thermal fluctuations at infinity, so the only variance is due to fluctuations in the location of the horizon, which are captured by the subleading logarithmic term discussed above.

The most obvious solution to this problem was discussed in [27], which is to normalize the charge relative to its average by $N$,

$$U \overset{?}{:=} \frac{1}{N}(H_B - \langle H_B \rangle). \tag{40}$$

This cancels the explicit $N$-dependence, so that $U$ has a well-defined large-$N$ limit. However, as in the case of asymptotically flat space considered in the previous section, this subtraction renormalization scheme again suffers from the issue that it removes the area contribution of interest, cf. the discussion below (21). Unlike the previous case however, the main issue is the large-$N$ limit, not the UV divergence *per se*. That is, recall from the general prescription in sec. 2 that we require an operator $X \in \mathfrak{A}'_B$. In [27], $X$ was taken to be (40) with $B \to B'$, which is well-defined as $N \to \infty$. Also recall however that there is an inherent ambiguity $X + c$ in the crossed product construction, manifesting in an additive constant $c$ in the final expression for the generalized entropy. In analogy with our flat space renormalization above, we may therefore absorb $\langle H_B \rangle / N$ into $c$, and work with

$$U := \frac{H_B}{N}. \tag{41}$$

---

[32]Technically, this will only be true if the region in question is sufficiently small so that the bulk wedge is not deformed by the black hole. Vacuum is then understood to mean that there are no excitations in the TFD state.

We note that this may be a very unwise choice of normalization if one wishes to do anything so practical as compute scattering amplitudes in the large-$N$ gauge theory, since $\langle U \rangle \sim N$. However, as explained above, it is important that we retain the vacuum expectation value of the modular charge for the region, so we must in any case use some renormalization scheme other than (40). The choice (41) simply shifts the entropy by the total energy, which scales like $N$.[33] In other words, one may choose whether to absorb this factor into the renormalization of $U$, or into the result for the entropy; but the former removes the area contribution of interest, so we choose the latter.

In the strict large-$N$ limit, $U$ is central, since it commutes with any element of the region $B$. That is, for any $A \in \mathfrak{A}_B$,

$$[U,A] = \frac{1}{N}[H_B,A] = -\frac{i}{N}\partial_t A, \tag{42}$$

which vanishes at $N \to \infty$. Consequently, $U$ commutes with $\mathfrak{A}_B$, and the dynamics are trivial, so if we adjoin $U$ to the type III algebra $\mathfrak{A}_B$, we obtain a simple tensor product

$$\widetilde{\mathfrak{A}}_B := \mathfrak{A}_B \otimes \mathfrak{A}_U, \tag{43}$$

where $\mathfrak{A}_U$ is the abelian algebra of bounded functions of $U$. Incidentally, the reason we have not used a subscript $B$ on the operator $U$ is that the same operator may be adjoined to the commutant via

$$\widetilde{\mathfrak{A}}_{B'} := \mathfrak{A}_{B'} \otimes \mathfrak{A}_U, \tag{44}$$

so that $\widetilde{\mathfrak{A}}_B$ and $\widetilde{\mathfrak{A}}_{B'}$ share the central generator $U$.[34] In this case however, the algebras are still type III, and the entropy remains ill-defined.

As shown in [27] for the TFD, the nature of these algebras radically changes if one works perturbatively in $N$, since the generator $U$ is no longer central as one backs away from the large-$N$ limit. The same is true for the subregions considered here. Recalling the general prescription in sec. 2, we wish to adjoin $H_B/N$ to the algebra of the region $B$. At finite $N$, this is a non-central element of $\mathfrak{A}_B$, and therefore generates an inner rather than outer automorphism of the algebra. Therefore, we instead take $T = H/N$ (which does generate an outer automorphism), and $X = H_{B'}/N$ (which belongs to the commutant, since $H/N \neq 0$ at finite $N$). We may then adjoin $T + X$ to $\mathfrak{A}_B$ to obtain the crossed product algebra[35]

$$\widehat{\mathfrak{A}}_B = \mathfrak{A}_B \rtimes \mathfrak{A}_{T+X}, \tag{45}$$

which is type II$_\infty$. Note that this effectively adjoins $U$, so that $\widehat{\mathfrak{A}}_B$ is the algebra of the region $B$ together with its associated modular flow. As in [27] however, this description is only perturbative in $1/N$; in particular, it reduces to a type I algebra for integer $N$.

While we have discussed this procedure for the boundary subregion $B$, subregion-subregion duality implies that the bulk dual of $\widehat{\mathfrak{A}}_B$ is the type II crossed product algebra $\widehat{\mathfrak{A}}_b$, constructed precisely as above by adjoining the bulk modular Hamiltonian $h/N$ to the initially type III algebra $\mathfrak{A}_b$ of the entanglement wedge $b$. In the case of vacuum, the entanglement wedge is isometric to AdS-Rindler, with the generator of the flow given by (37).

This implies that at least for the class of separable states discussed in sec. 2, one may define density matrices $\widehat{\rho}_b$ for this new type II algebra of the entanglement wedge, whose entropy is

$$\begin{aligned} S(\widehat{\rho}_b) &= \langle \widehat{\Psi}|h_b|\widehat{\Psi}\rangle - \langle \widehat{\Psi}|\ln g(h_b)|\widehat{\Psi}\rangle \\ &= \frac{A(\Sigma)}{4G} - \langle \widehat{\Psi}|\ln g(h_b)|\widehat{\Psi}\rangle, \end{aligned} \tag{46}$$

---

[33]We note that the constant $c$ also diverges as $G \to 0$ in [34]. In this case, we may give this divergence a natural interpretation as the total energy associated with $N \to \infty$ degrees of freedom.

[34]That $U$ is the same in both algebras follows from the fact that for any $U' := H_{B'}/N$, the difference $U' - U = H/N \to 0$ as $N \to \infty$. See [27] for more detailed discussion.

[35]Note that the component $\mathfrak{A}_{T+X}$ was simply denoted $\mathfrak{A}_{\mathbb{R}}$ in sections 2 and 3, but we have explicitly indicated that this consists of bounded functions of $T + X$ here to avoid confusion with the commutant below.

where $A(\Sigma)$ is the area of the entangling surface, in this case the AdS-Rindler horizon. See [79] for a recent discussion of the relation $\langle h_b \rangle = A(\Sigma)/4G$ for AdS-Rindler. Of course, as mentioned in the previous section, the area of the entangling surface is infinite due to the infinite distance to the horizon. However, this divergence is reasonably well-understood, and can be regulated with an IR cutoff, which is dual to the UV cutoff on the boundary. Indeed, this is analogous to the exchange of IR/UV cutoffs in the mapping from the Rindler wedge to the ball-shaped region in the CFT discussed in subsec. A; see for example [55, 81]. Thus, while these familiar divergences still remain, the crossed product construction allows one to define density matrices and traces for a large class of operators which would not be possible in the original type $\mathrm{III}_1$ algebra of the subregion. It would be interesting to have a description of theories that is free of divergences completely, which presumably would connect with the type I algebras appearing for certain black holes in string theory. In those cases, the entropy can be tied to an explicit counting of the quantum gravitational microstates making up the black hole, while in the present case it is a measure of ignorance of what lies beyond the entangling surface.

We note that the algebra of operators in AdS-Rindler in the large-$N$ limit has been discussed very recently in this context in [82]. The main conceptual distinction is our emphasis that this construction relies on adjoining the modular Hamiltonian to the algebra of the entanglement wedge, and hence in principle applies in complete generality, though as mentioned above the expression (37) only holds for the simple case when the entanglement and causal wedges coincide. At a technical level, [82] is primarily concerned with the properties of the type III algebra $\mathfrak{A}_b$, and also discusses the regulation of the infinite area surface in AdS mentioned in the previous paragraph, while we are primarily interested in the construction of the type II crossed product algebra and the associated generalized entropy.

In summary, adjoining the (suitably normalized) modular Hamiltonian to the subalgebra of observables in the entanglement wedge reduces the algebra to type II perturbatively in $1/N$, and similarly for the corresponding boundary subalgebra in the large-$N$ gauge theory. In the vacuum state, where the flow is geometric, the modular Hamiltonian admits a local expression in terms of the AdS-Rindler boost generator (37). However, the basic construction still holds away from vacuum, when the entanglement and causal wedges no longer coincide. In the next two subsections, we comment on the cases in which this occurs, namely excited states and disjoint regions on the boundary.

## 4.2 Excited states

In the previous subsection, we considered the crossed product algebra for an arbitrary entanglement wedge in the vacuum state, which is isometric to the AdS-Rindler wedge. Away from vacuum however, the causal and entanglement wedges will differ, since the presence of matter shifts the extremal surface further into the bulk. In the boundary, the lack of gravity means that the causal diamond $D$ will remain geometrically unchanged, but the form of the modular Hamiltonian will reflect the excited state in the CFT. The latter scenario has been considered by [83] (see also [84]), for states in $\mathfrak{A}_B$ of the form

$$\rho_B = \rho_{B,0} + \delta\rho \,, \tag{47}$$

where $\rho_{B,0}$ is the density matrix for the region $B$ in the vacuum state, and $\delta\rho$ is a perturbation representing the excitation due to the action of some primary operators. Of course, for the type III algebra $B$, none of these objects exist, so this expansion must be considered purely formal. However, we could consider states in $\widehat{\mathfrak{A}}_B$ of the form

$$\widehat{\rho}_B = \widehat{\rho}_{B,0} + \delta\widehat{\rho} \,, \tag{48}$$

where $\widehat{\rho}_{B,0}$ is the vacuum state of the type II algebra constructed in the previous subsection, and $\delta\widehat{\rho}$ is a perturbation caused by the action of a primary operator on $\mathfrak{A}_B$ (which acts trivially

on $L^2(\mathbb{R})$).[36] Formally, we would then expect the analysis of [83] to go through unchanged. In particular, their central result is that the generator of the modular flow for the excited state of $B$ is given by

$$K_B = K_{B,0} + \sum_{n=1}^{\infty} (-1)^n \delta K_n \,, \tag{49}$$

where $K_{B,0}$ is the one-sided charge for vacuum state reduced to the region, and the corrections $\delta K_n$ are given explicitly in [83]. This expression is again purely formal, since the modular Hamiltonian has support on both the region and the complement,

$$K = K_B - K_{B'} \,, \tag{50}$$

where we again assume Haag duality. Note that this is *not* equal to (37), and that the causal (i.e., AdS-Rindler) wedge is not preserved under the modular flow generated by $K$. Additionally, if the CFT is a large-$N$ gauge theory, then additional care is needed to ensure a well-defined limit as $N \to \infty$. Conceptually however, the analysis in the previous subsection still applies. If the perturbative analysis above holds, it would enable one to express the entropy of the region as in (46), with additional subleading corrections from the sum over $\delta K_n$.

However, we emphasize that in any case, the area term $A(\Sigma)$ in (46) is the area of the entangling surface that defines the region. By the quantum extremal surface prescription [24], or by the more abstract algebraic arguments given above, this is determined self-consistently for a given background state, including all matter contributions which backreact on the geometry and hence change the location of the surface. Thus, if one considers an entanglement wedge in the global AdS vacuum, $A(\Sigma)$ will be the area of the surface in vacuum; if one considers an entanglement wedge in some highly excited state, then $A(\Sigma)$ will be the (generically different) area of the deformed entangling surface in that state; see fig. 2.[37] The existence of an expression like (49) would merely allow one to write the latter in terms of a perturbation about the former. This is the reason we put "excited state" in quotes below (21), since the state in which the modular charge is defined is by construction the cyclic and separating state (i.e., the "vacuum") for the algebra $\mathfrak{A}_b$. We illustrate this in fig. 2.

From this perspective, the fact that the generalized entropy of excited states was the primary object of consideration in previous works is slightly strange, since the area is fixed by the subregion, i.e., by the cyclic and separating (vacuum) state for the subalgebra. This is perhaps an artefact of the use of the subtraction scheme to normalize the modular charge, which removes the vacuum contribution, leading to the argument from relative entropy mentioned above. One could of course consider trivial excitations around the vacuum state, i.e., those that have no impact on the location or shape of the entangling surface, but any significant excitation must necessarily correspond to a different subregion and hence a different subalgebra.[38] Nonetheless, suppose one wishes to consider the entropy of such an insertion, such that the area $A(\Sigma)$ is unchanged. Then this will introduce a difference in type III von Neumann entropies between the two states in the expression for the generalized entropy, and this difference is well-defined.

To see this, let us recall part of the argument from Wall [51], namely the relationship between the relative entropy and von Neumann entropy. This was not proven, insofar as it

---

[36]Note that what we are calling $\widehat{\rho}_{B,0}$ here was called $\widehat{\rho}_B$ in the previous subsection, but we have again overloaded the notation to avoid cluttering the manuscript with excessive subscripts.

[37]The reader may wonder how the duality between bulk and boundary algebras is maintained for the case when bulk region is enlarged. The answer is that when we speak of the algebra of observables here, we have in mind the GNS representation of the abstract algebra in a particular state (as a linear functional in the abstract algebra, as opposed to a vector in a particular Hilbert space). The abstract algebra does not change when moving to an excited state, but the Hilbert-space representations $\Psi$ and $\Phi$ may differ.

[38]Generically, one will simply be an inclusion of the other.

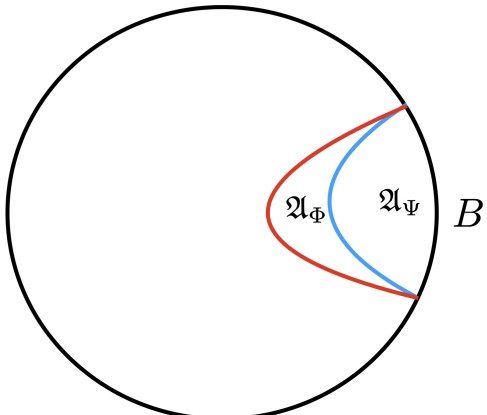

Figure 2: Restricting to a subregion $B$ on the boundary, the algebras of observables in the bulk are illustrated relative to a particular state in the Hilbert space. Here $\Psi$ represents the global AdS vacuum, and $\Phi$ some non-trivial excited state (e.g., a black hole deep in the bulk, or some collection of matter fields). The backreaction due to the excited state deforms the entangling surface deeper into the bulk, so that the algebra $\mathfrak{A}_\Phi$ is an inclusion of $\mathfrak{A}_\Psi$. Thus, considering the entropy of an excited state in the crossed product construction is formally identical to considering the entropy of the vacuum state, since in either case the expectation value of the modular charge returns the correct area contribution. Only in the case of trivial excitations, e.g., very light insertions of operators in $\mathfrak{A}_\Psi$ or $\mathfrak{A}_\Phi$ which do not alter the location of the corresponding entangling surfaces, does it make sense to consider "excited states" via the relative entropy, in which case we show that this simply gives an additional contribution from the difference in type III entropies.

assumes a suitable renormalization scheme for the modular Hamiltonian and von Neumann entropy in the type III algebra. However, it can be applied more rigorously to the type II system, in which these objects are well-defined. Accordingly, let $\widehat{\Phi}$ denote a trivially excited state relative to the type II vacuum $\widehat{\Psi}$. Again, by trivially excited, we mean that it does not change the area of the extremal surface used in defining the region, and by extension the local vacuum state. (Note that the state denoted $\Phi$ in fig. 2 is not a trivial excitation, and is not the state $\Phi$ discussed here.) Then the relative entropy may be written

$$
\begin{aligned}
S(\widehat{\Phi}||\widehat{\Psi}) &= -S(\widehat{\Phi}) - \text{tr}(\rho_{\widehat{\Phi}} \ln \rho_{\widehat{\Psi}}) \\
&= -S(\widehat{\Phi}) - \langle \widehat{\Phi}| \ln \rho_{\widehat{\Psi}} |\widehat{\Phi} \rangle \\
&= -S(\widehat{\Phi}) + \langle \widehat{\Phi}| \beta K |\widehat{\Phi} \rangle - \langle \widehat{\Phi}| \ln g(K) |\widehat{\Phi} \rangle \,,
\end{aligned}
\tag{51}
$$

and therefore,

$$
S(\widehat{\Phi}) = \beta \langle K \rangle_{\widehat{\Phi}} - S(\widehat{\Phi}||\widehat{\Psi}) - \langle \ln g(K) \rangle_{\widehat{\Phi}} \,,
\tag{52}
$$

where $K$ is the modular charge for the state $\widehat{\Psi}$, i.e.,

$$
\rho_{\widehat{\Psi}} = e^{-\beta K} g(K) \,.
\tag{53}
$$

Note that since these are type II expressions, each individual term is well-defined. Since the $\ln g$ term is the universal contribution discussed above, we suppress it henceforth.

Suppose now we take the expression from eqn. (42) in [51] for the relative entropy of states in the type III theory as a difference in free energies,

$$
S(\Phi||\Psi) = \beta \langle K \rangle_\Phi - S_\Phi - \beta \langle K \rangle_\Psi + S_\Psi \,.
\tag{54}
$$

Naïvely, this expression is purely formal, since the von Neumann entropies on the right-hand side are not defined. However, the particular difference that appears here is in fact UV finite. One way to see this is to realize that relative entropy is well-defined even in type III theories, and is in fact equal to the relative entropy of the corresponding type II states, i.e., $S(\Phi\|\Psi) = S(\widehat{\Phi}\|\widehat{\Psi})$. Hence, substituting (54) into (52), we obtain

$$S(\widehat{\Phi}) = \beta \langle K \rangle_\Psi + S_\Phi - S_\Psi - \langle \ln g(K) \rangle_{\widehat{\Phi}}, \tag{55}$$

where we used the fact that the expectation value of the modular charge agrees for both type III and type II states, i.e.,

$$\langle \widehat{\Psi}|K|\widehat{\Psi}\rangle = \int_{-\infty}^{\infty} dK' \, g(K') \langle \Psi|K|\Psi\rangle = \langle \Psi|K|\Psi\rangle, \tag{56}$$

where we used the fact that $g^{1/2} \in L^2(\mathbb{R})$, cf. (24). Of course, the derivation above is more convoluted than necessary for the sake of connecting with familiar expressions in the literature: since $\Phi$ here is a trivial excitation in the algebra $\mathfrak{A}_\Psi$, $\langle K \rangle_\Phi = \langle K \rangle_\Psi$. One can thus consider the right-hand side of (54) to be simply the difference in entropies, and the first terms on the right-hand sides of (52) and (55) are identical.

Thus we see that for small excitations in our subalgebra, the generalized entropy picks up a bulk contribution given by the difference in von Neumann entropies of the excited and vacuum states. Additionally, since the left-hand side and the first term of the right-hand side of (55) are well-defined, the combination of type III quantities $S(\Phi) - S(\Psi)$ must be well-defined as well. We emphasize again however that the excitation does not give a separate area term, since this is derived from the state of the algebra localized to the subregion.

## 4.3 Disjoint boundary subregions

The other case in which the casual and entanglement wedges are distinct, even in the vacuum state, is if the boundary region $B$ consists of disjoint regions $B_i$,

$$B = \bigsqcup_{i=1}^{n} B_i, \tag{57}$$

such that the spatial extent of all $B_i$ collectively exceeds half the boundary. As discussed above, the reason for this is that the entanglement wedge exhibits the switchover effect, but the causal wedge does not; see fig. 1.

For concreteness, consider the case $n = 2$. Formally, one may write the generator for the modular flow of $B = B_1 \bigsqcup B_2$ as [85]

$$K = K_1 + K_2 - K_{12}, \tag{58}$$

where $K_i$ is the generator for $B_i$, and $K_{12}$ is a non-local contribution related to the mutual information[39]

$$I(B_1, B_2) \geq \frac{(\langle \mathcal{O}_1 \mathcal{O}_2 \rangle - \langle \mathcal{O}_1 \rangle \langle \mathcal{O}_2 \rangle)^2}{2\|\mathcal{O}_1\|^2 \|\mathcal{O}_2\|^2}, \tag{59}$$

where $\mathcal{O}_i$ is a bounded operator with support in $B_i$, and $\|\cdot\|$ is the operator norm. In this expression, the expectation value is computed in the state $\rho_B$ (or rather, $\widehat{\rho}_B$). Naïvely, we would expect that for disjoint regions, we could formally decompose the state as

$$\rho_B = \rho_1 \otimes \mathbb{1}_2 + \mathbb{1}_1 \otimes \rho_2, \tag{60}$$

---

[39]This expression for the mutual information follows from Pinkser's inequality and the definition $I(A, B) := S(\rho_{AB}|\rho_A \otimes \rho_B)$.

in which case

$$\langle \mathcal{O}_1 \mathcal{O}_2 \rangle = \langle \mathcal{O}_1 \rangle_1 \langle \mathcal{O}_2 \rangle_2 \,, \tag{61}$$

and hence the mutual information vanishes. This suggests that, below the switchover point

$$K = K_1 + K_2 \,, \tag{62}$$

which reflects the fact that, in the bulk, the two entanglement wedges are spacelike separated, so the respective modular flows should be given by the independent vacuum or excited state expressions above; this is illustrated in the left panel of fig. 1.

Above the switchover point however, the bulk entanglement wedges merge into the single connected region shown in the right panel of fig. 1. In this case, the mixing of bulk operators implies that the boundary factorization implicit in (61) cannot hold without breaking the duality between the bulk and boundary subalgebras. We are thus lead to the conjecture that the CFT dual of the switchover in the bulk is the non-vanishing of the mutual information[40] between the disjoint boundary subregions. In this case, $K_{12}$ will take a finite value, thereby lowering the area contribution $A(\Sigma)/4$ to the entropy from $K$, as must happen from the bulk perspective, since the maximum entropy occurs when $B_1 \bigsqcup B_2$ encompasses exactly half the boundary (in the vacuum state).

The above picture is a slight oversimplification, because the algebra may contain bilocal operators with support in both subregions. However, we would still expect that below the switchover point, the modular flow of such operators does not mix between $B_1$ and $B_2$. Indeed, a more operational diagnostic of this switchover in the boundary are based on the intuition that after the switchover, the boundary modular flows should mix operators between the two disjoint subregions. For example, take two operators $\mathcal{O}_1$ localized to the entanglement wedge dual to the boundary subregion $B_1$, and $\mathcal{O}_2$ localized to the entanglement wedge dual to $B_2$. Below the switchover point,

$$[\mathcal{O}_1, \mathcal{O}_2] = 0 \,. \tag{63}$$

Now act on both operators with the modular Hamiltonian for $\mathfrak{A}_B$, and let $s$ parametrize the resulting modular flow. Then there should exist a critical value of $s_0$ at which[41]

$$[\mathcal{O}_1(s_0), \mathcal{O}_2(s_0)] \neq 0 \,, \tag{64}$$

indicating that the modular flow has transported the support of the operator $\mathcal{O}_2$ into the interior region covered after the switchover, cf. fig. 1. Again however, this will not necessarily be true for all operators, and hence it is an interesting open question to find appropriate operators in the CFT that are sensitive to the breakdown of geometricity of the flow in the bulk. It is interesting to speculate that there may be a connection between the sensitivity to non-geometricity of the modular flow and exponential complexity of boundary operators, since in some sense the newly accessible region in the entanglement wedge after the switchover resembles the black hole interior from the perspective of the causal wedge.

## 5  Discussion

In this work, we have shown that the crossed product construction introduced for the TFD in [27], and shortly thereafter applied at the level of the global type III algebra of particular

---

[40]The mutual information is upper bounded by the entanglement of purification. The latter is conjectured to be dual to the entanglement wedge cross-section, which exhibits similar behavior to the mutual information in the context of the entanglement wedge phase transition [86].

[41]This is superficially similar to the critical value identified in [29] indicating the existence of the horizon in the TFD. However, the modular Hamiltonian does not evolve operators beyond the horizon, so this cannot actually be seen with a consistent choice of sign for the modular inclusions of the exterior algebras [28, 87].

theories in [34–36], can be applied to in principle arbitrary subregions of quantum field theories by adjoining the appropriate modular Hamiltonian. This allows one to associate a type II algebra of observables to local spacetime regions (which are otherwise of type III), admitting a trace and therefore a well-defined notion of entropy. In this sense, the construction provides a refinement of the association of observable algebras to spacetime regions that forms the basis of the Haag-Kaster axioms, in the sense that it considers the type III algebra together with its modular automorphism group. The von Neumann entropy of the crossed product algebra is naturally associated to the generalized entropy of the subregion in which it localizes.

The simplest example is that of Rindler space, where the modular Hamiltonian is the Lorentz boost generator for both the left and right wedges. Adjoining the modular Hamiltonian to the algebra in this case strengthens the connection between modular time and horizon entropy encoded in the KMS condition. If the QFT in question is a CFT, then the Rindler wedge may be conformally mapped to an open ball in asymptotically flat space, which allows us to immediately repeat the analysis in the latter case. This lays the groundwork for the application to holography, since the causal diamond of an open ball in the boundary CFT is dual to the entanglement wedge in bulk AdS space. In all cases, one obtains a well-defined formula for the von Neumann entropy of the subsystem, which reproduces the leading area term due to entanglement across the horizon or extremal surface, as well as the subleading logarithmic term due to thermal fluctuations.

Additionally, we have provided a simple but novel argument for the fact that the entanglement wedge is the appropriate bulk dual of a boundary subregion.[42] This relies on the duality between the bulk and boundary von Neumann algebras associated to the region, initially employed in the holographic context in [28] (inspired in large part by earlier work by Papadodimas and Raju, Jafferis, and others [64,65,90]) and recently popularized in [63] (see also [80]). In particular, modular theory provides a natural isomorphism between the algebra of a subregion $\mathfrak{A}$ and its commutant $\mathfrak{A}'$, which provides a foundational explanation for the switchover effect that characterizes bulk extremal surfaces. In the presence of gravitational degrees of freedom, not all subregions are viable for the definition of gauge-invariant algebras of observables, but subregions bounded by extremal surfaces are. While it seems obvious that the minimality requirement imposed on the selection of the entangling surface in the literature follows automatically, we are not aware of an argument that would prevent a scenario in which the minimal surface is in fact not consistent with the algebraic isomorphism, which would therefore select a different, non-minimal surface as the appropriate demarcation of the bulk region. It would be very interesting to explore the role of the isomorphism constraint in more detail, as well as the connections with mutual information discussed in the previous section.

The algebraic approach also provides a basis for a more thorough treatment of the generalized second law of thermodynamics. As remarked previously, the natural entropy on the crossed product algebras coincides with the generalized entropy of the underlying subregion. Given an inclusion of such algebras arising from a subregion inclusion, the generalized second law follows from the simple fact that the entropy cannot decrease under restriction to a subalgebra as remarked speculatively in [35] and proven for the type III case under the assumption of some suitable renormalization scheme in [51]. Our extension of this formalism to arbitrary subregions is also exciting in the context of modular inclusions in holography. These were originally introduced by Wiesbrock [91] as a means towards understanding the physical interpretation of the modular operator and modular conjugation in Tomita-Takesaki theory, and were first applied to AdS/CFT in the particular case of a double-trace deformation of the TFD in [28], where it was proposed that modular inclusions may provide a useful means of probing the black hole interior. Several years later, this idea was developed in detail in [63], which

---

[42]For related discussions of von Neumann algebras associated to holographic subregions, see [88,89].

also highlighted the emergent type III$_1$ factor in the boundary CFT above the Hawking-Page transition.[43] We hope to revisit this topic in the context of the type II crossed product algebras in future work.

While the notion of modular flow has appeared in the AdS/CFT literature before (see for example [64, 65, 92]), most works have used the term "modular Hamiltonian" to mean $K = -\ln \rho$ for an arbitrary reduced density matrix $\rho$. As we have emphasized, this is not technically correct, as the modular Hamiltonian has support on both the region and its commutant, and furthermore cannot be factorized for the type III theories of greatest interest. Nonetheless, these and other works provided important advancements in understanding the relation between entropy and area in holography. In particular, a diverse body of work (for a criminally short sample, see [74, 93, 94]) seems to suggest that the bulk spacetime emerges from entanglement in the boundary theory, encapsulated in the phrase "it from qubit".[44] Attempting to understand this however will require going beyond the current approach, which is based on the application of type I reasoning to type III systems, in particular assuming the existence of a density matrix and a trace. As some authors have pointed out [17, 26, 28], this is problematic for a variety of reasons, and many of the most vexing puzzles in quantum gravity – such as the emergence of spacetime or the black hole information paradox – will almost certainly require a more fundamental treatment of these issues. The recent interest in the application of ideas from operator algebras and in particular the crossed product construction represents an extremely exciting development towards a deeper understanding of entanglement in quantum field theories and holography in particular, and we look forward to future work in this direction.

## Acknowledgments

S.A.A would like to thank Alexander R. H. Smith, Qidong Xu, Jackson Yant, and Miles Blencowe for interesting conversations on entanglement in quantum field theory and gravity. R.J. would like to thank Ibra Akal, Umut Gürsoy, and Nima Lashkari for stimulating discussions on von Neumann algebras in QFT and holography, as well as the organizer, Watse Sybesma, and participants of the Gravity Sagas conference in Reykjavík, Iceland.

## A   Mapping the Rindler wedge to an open ball

The map from the right Rindler wedge to an open ball in a CFT (29) was used by [55] to determine the modular Hamiltonian (32) for the domain of dependence $D$ of the latter. Since this map is fundamental to our results, and is used in the main text to also map the left Rindler wedge to $D'$, we review it here, and elaborate on some aspects left implicit in the original paper. We also note that the exposition in (version 2 of) [55] contains a sign error: the shift vector $C^\mu$, defined below their (2.11), should instead read

$$C^\mu = \left( 0, -\frac{1}{2R}, 0, \dots, 0 \right). \tag{A.1}$$

A simple way to see this is as follows:  a special conformal transformation (SCT) maps

---

[43]The type III nature of these algebras was assumed in [28] based on the idea of subregion-subregion duality discussed here, but no argument was given for the specific case when the subregion consists of the entire boundary. See however [32].

[44]This is a modern version of the phrase "it from bit" coined by Wheeler in [95].

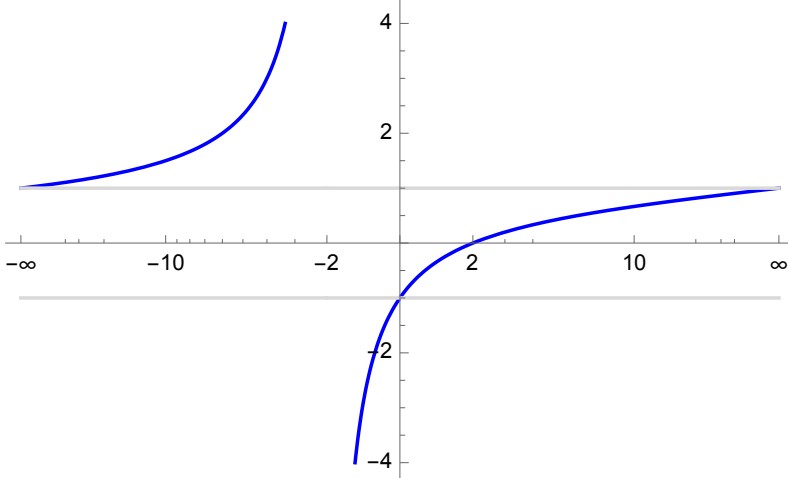

Figure 3: Plot of the $\mu = 1$ component of the SCT (29) with $C^1 = -\frac{1}{2R}$. We have set $R = 1$ for visualization purposes. The horizontal axis is the original Rindler coordinate $X^1$, and the vertical axis is the transformed coordinate $x^1$. Note that the entire right Rindler wedge, represented by the region $X^1 > 0$, is mapped to the interval $-R < x^1 < R$ (indicated by the gray horizonal lines), while the discontinuity at $X^1 = -2R$ splits the left Rindler wedge over the complement.

Minkowski coordinates $X^\mu$ to [96]

$$x^\mu = \frac{X^\mu - C^\mu X^2}{1 - 2C \cdot X + C^2 X^2}, \tag{A.2}$$

and can be equivalently expressed as an inversion, then a translation by $-C^\mu$, followed by another inversion:

$$\frac{x^\mu}{x^2} = \frac{X^\mu}{X^2} - C^\mu. \tag{A.3}$$

Now, specifying to $1+1$ dimensions for illustrative purposes, consider the point $X^\mu = (0, 2R)$. Had we chosen $C^\mu = (0, \frac{1}{2R})$ as in [55], then under the SCT above, the spatial component $X^1$ would be inverted to $\frac{1}{2R}$, then shifted to 0, and finally inverted to $\infty$. This is unavoidable: there will always be a singularity at $\frac{1}{C^1}$. However, this implies that the transformation (2.20) in [55] cannot possibly map the right Rindler wedge to the ball with radius $R$. The solution is of course to shift the singularity into the left Rindler wedge, which is accomplished by choosing $C^1 = -\frac{1}{2R}$. In this case, the point $X^1 = \infty$ is inverted to 0, shifted to $\frac{1}{2R}$, and then inverted to $2R$. The final shift by $2R^2 C^\mu$ in (29) then reduces this to $R$, ensuring that the entire right wedge is mapped to the ball. See fig. 3 for an illustration in $1+1$ dimensions.

This sign error propagates though some of the subsequent expressions of [55]; in particular, the corrected version of their (2.13) reads

$$x^\pm(s) = R \frac{(x_\pm \pm R) + e^{2\pi s}(x_\pm \mp R)}{(R \pm x_\pm) + e^{2\pi s}(R \mp x_\pm)}, \tag{A.4}$$

and the infinitesimal shift of the temporal $x^0$ and transverse $r$ coordinates in their (2.18) becomes

$$\delta x^0 = 2\pi \frac{r^2 - R^2}{2R} \delta s, \qquad \delta r = 0. \tag{A.5}$$

However, their final result for the modular operator in (2.20) is nonetheless correct. To verify this, let us review the core argument in [55]. Denoting the unitary operator that generates the

modular flow in the diamond region by $U_D(s) = e^{isH_D}$, the action on spinless primary fields in the CFT is given by

$$U_D(s)\phi(x[s_0])U_D(-s) = \Omega(x[s_0])^\Delta \Omega(x[s_0 + s])^{-\Delta}\phi(x[s_0 + s]), \tag{A.6}$$

cf. (2.17) in [55], where $x[s_0]$ indicates that the position is considered a function of the flow parameter $s$, starting at $s_0$. We then consider the action of an infinitesimal flow on the surface $x^0 = 0$. As argued in [55], the shift in the conformal prefactor $\Omega$ vanishes to linear order in $\delta s$. Therefore, the transformed modular flow induces the following infinitesimal change in the fields:

$$\phi(x[s_0 + s]) \approx \phi(x[s_0] + s\partial_s x[s_0]) \approx \phi(x[s_0]) + s\partial_s x[s_0]\partial_x\phi(x[s_0])). \tag{A.7}$$

Since, by (A.5) the linear variation in the transverse directions vanishes, we need only consider the variation in the temporal direction $x^0$. From the definition of $U_D(s)$ above, $H_D$ is the generator of this transformation, which acts by conjugation on the fields:

$$U(s)\phi(x)U(-s) = e^{isH_D}\phi(x)e^{-isH_D} = \phi(x) + is[H_D, \phi(x)] + \mathcal{O}(s^2). \tag{A.8}$$

Comparing the linear order terms in (A.7) and (A.8), we thus identify

$$[H_D, \phi(x)] = -i\partial_s x^0\,\partial_0\phi(x) = 2\pi i\frac{R^2 - r^2}{2R}\,\partial_0\phi(x), \tag{A.9}$$

where in the second step we have used (A.5), and we have suppressed the dependence of $x^\mu$ on $s_0$ for compactness. Let us now take the form of the modular operator given by [55] as an ansatz,[45]

$$H_D = 2\pi \int_V \mathrm{d}^d x\, \frac{R^2 - r^2}{2R} T^{00}(x) + c, \tag{A.10}$$

where $c$ is some constant to fix the normalization of the trace,[46] and $V$ denotes the interior of the ball. Substituting (A.10) into the commutator, we have

$$[H_D, \phi(x)] = 2\pi \int_V \mathrm{d}^d y\, \frac{R^2 - r^2}{2R}[T^{00}(y), \phi(x)]. \tag{A.12}$$

Now, recall that the stress tensor is the conserved current associated to translation symmetry; by Noether's theorem, we may write this as

$$T^{\mu\nu} = \frac{\delta\mathcal{L}}{\delta(\partial_\mu\phi)}\partial^\nu\phi - \eta^{\mu\nu}\mathcal{L}, \tag{A.13}$$

where the metric is $\eta_{\mu\nu} = \mathrm{diag}(-1, 1, \ldots, 1)$. Thus,

$$[T^{00}(y), \phi(x)] = \left[\phi(x), \frac{\delta\mathcal{L}}{\delta(\partial_0\phi)}\partial_0\phi(y) - \mathcal{L}\right] = i\delta(x - y)\partial_0\phi(x), \tag{A.14}$$

---

[45]Note that we use the modern convention in which the spacetime dimension is $D = d + 1$, while in [55] the spacetime dimension is denoted $d$.

[46]Since the trace in the type II theories of interest are only defined up to a constant rescaling, we simply drop the constant. It is relevant however in comparing different derivations of this expression. In particular, [55] work with the traceless stress tensor for a massless scalar field, which is

$$T_{\mu\nu} = \partial_\mu\phi\partial_\nu\phi - \frac{1}{2}g_{\mu\nu}(\partial\phi)^2 + \frac{D-2}{4(D-1)}(g_{\mu\nu}\Box\phi^2 - \partial_\mu\partial_\nu\phi^2). \tag{A.11}$$

One can verify that $T^\mu_\mu = 0$ on-shell. In contrast, [57] instead use the usual stress tensor which includes only the first two terms. Consequently the result of the latter for the generator of the modular flow contains an extra term relative to the expression (A.10).

where we used the canonical commutation relation

$$\left[\phi(x), \frac{\delta\mathcal{L}}{\delta(\partial_0\phi)}\right] = i\delta(x-y).$$ (A.15)

Substituting (A.14) into (A.12) then yields

$$[H_D, \phi(x)] = 2\pi i \frac{R^2 - r^2}{2R} \partial_0\phi(x),$$ (A.16)

which is (A.9), as desired.

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
