# Peer review of "Crossed product algebras and generalized entropy for subregions"

_SciPost Physics, doi:SciPost Phys. Core 7, 020 (2024)_

## Round 1 · Referee Report · Anonymous (Referee 1) · 2023-10-11

Strengths

1- The construction studied by the authors is very natural and general and has potential applications also outside of high-energy physics.

2- The authors consider many physically relevant scenarios.

Weaknesses

1- The paper is at time too terse, omitting some details which could be helpful to the non-expert reader.

Report

In a spatial subregion, the algebra of observables does not possess neither a well defined trace nor a density matrix, leading to ultraviolet divergences in the entanglement entropy.
In this work, the authors review and extend a construction that alleviate this problem, the crossed product construction.
Every algebra of observables possesses a group of automorphisms generated by the modular Hamiltonian. By adjoining the action of this group to the algebra, the resulting extended algebra admits a trace and a density matrix. This allows the authors to define a generalised entanglement entropy which is in principle free from UV divergences.

The crossed product construction was already introduced in previous works, in particular [Witten 2021], where it was applied to the thermofield double state. The authors review this construction and extend it to many physically relevant examples. The main novelty with respect to the previous works is that the authors show that the construction is in principle independent of gravity, since it only requires the modular Hamiltonian.

After reviewing the general procedure, the authors apply it to several situations, in particular they consider the vacuum of a relativistic QFT in a Rindler wedge and the one of a CFT in a ball shaped region.
The authors consider also examples in holographic theories. They first study the vacuum state in the AdS/Rindler region, in which the action of the modular Hamiltonian is geometric, and then the case of excited states and of multiple boundary intervals.

The construction studied is very natural and general. While the procedure was already introduced in the literature, the authors demonstrate convincingly that it is independent of gravity. For this reason, this work has potentially vast applicability, even possibly in low-energy physics.

The authors put the paper in the appropriate context, citing the previous works and remarking the main differences between their results and the state of the art.

The paper is well written, however at time the authors omit some details that could be helpful to the non-expert reader. As an example, the modular Hamiltonian used in the construction depends on a specific state, however surprisingly the resulting extended algebra is independent of this state. This was well explained in [Witten 2021], but it is not very clear from this paper.

I recommend the publication of the paper after the authors answer some comments.

1- In Eq. (2.7), the authors show that the states belonging to the extended algebra are given by the tensor product of a state in the original algebra with a function $g$. This function $g$ enters the final result of the generalised entropy in, e.g., Eq. (3.14) for the Rindler wedge and Eq. (3.21) for the ball-shaped region.
This is analogous to the result of [Witten 2021], where the function $g$ was determined to be a Gaussian for the thermofield double state.
In this paper, however, the authors do not discuss how to determine $g$ in the physical situations they consider. In particular, it is unclear if $g$ is fixed or it remains free. The authors could comment on this function.

2- In Eq. (3.22) the authors consider the known result for the entanglement entropy of the vacuum of a 2d-CFT in an interval of length $\ell$ \[S = \frac{c}{3}\log\frac{\ell}{a}.\]The authors claim that the parameter $a$ plays the role of an infrared cut-off. This may simply be a typo, but $a$ is actually an ultraviolet cut-off on small lengths.
A way to see that this is the case is the construction of [Cardy, Tonni 2016]. In that paper, $a \ll \ell$ appears as the diameter of two small circles that have to be removed from the boundary of the interval. This regularisation is needed precisely because the entanglement entropy is UV divergent.
This is a minor point, but it is relevant to the stated purpose of comparing the known Eq. (3.22) with the authors' result in Eq. (3.21), since the generalised entropy (3.21) should be UV finite, differently from (3.22).

3- This last comment is optional, but i hope that the authors may clarify a doubt that arises with respect to excited states in AdS/CFT holography.
At pgs. 24-25 the authors discuss the case of excited states in an holographic large-$N$ CFT. They argue that by considering an excited state (keeping fixed the boundary region), the bulk dual region is modified in such a way as to include the bulk dual relative to the vacuum. As a consequence, the bulk algebra relative to the excited states is larger and contains the one relative to the vacuum.
My doubt regards the boundary algebra. Since the boundary subregion is kept fixed, one could naively expect that the boundary algebra does not change. Therefore it seems that the duality between boundary and bulk algebras does not hold anymore. I hope that the authors could clarify this point.

Requested changes

1- Comment on the role of the function $g$, by either briefly explaing how one could determine it or by making it explicit that it is left free.

2- Under Eq. (3.22), change "$a$ is an infrared cutoff" with "$a$ is an ultraviolet cutoff".

---

## Round 1 · Referee Report · Anonymous (Referee 2) · 2023-12-18

# Report: Crossed product algebras and generalized entropy for subregions

Dear Editor,

This manuscript discusses a generalization of a recent discussion of crossed products in gravity algebras to the Rindler space. The authors claim *"the crossed product construction represents a refinement of Haag's assignment of nets of observable algebras to spacetime regions by providing a natural construction of a type II factor"*. I have several conceptual comments and questions about the draft.

1. Page 1: *While any open spacetime region may be assigned a von Neumann algebra A(O), imposing further axioms on the assignment (1.1) leads to relations between the algebras defined on different regions*

   I find this statement misleading. We only associate $C^*$-algebras to open sets in spacetime. I do not know of any ways to assign a von Neumann algebra to an arbitrary open set in spacetime without taking the double commutant that, in local QFT with timeslice axiom, enlarges the region to its causal completion.

   As the authors elaborate: For a topologically trivial region, in many quantum field theories, in the standard KMS state, one proves Haag's duality. Taking the double commutant results in von Neumann algebras that are associated with double cones, or causally complete regions. For example, see the "time-like tube theorem".

2. Page 2: *These infinities arise from the lack of a finite trace on the algebras of observables, thereby obstructing the existence of density states localized to the region. This is due to the fact that the algebra of operators in type III theories does not admit a tensor factorization, in contrast to type I factors appearing in quantum mechanics*

   This statement is correct but misleading. Note that the issue of the existence of trace and tensor factorization are not to be confused. Type $II_1$ algebras have finite trace but still do not admit a reducible presentation on $B(\mathcal{H})$, required for tensor factorization.

3. Page 3: *In stark contrast to type III algebras, type II algebras do admit a well-defined trace, and hence a meaningful definition of von Neumann entropy for density states.*

   It is important to mention that the von Neumann entropy of type II algebras are still different in nature from type I entropy. It is crucial to remember that the type II$_\infty$ entropy is not derived from counting states, as opposed to the case of type I algebras.

4. Page 3: *The issue with this in the type III$_1$ algebra is two-fold: there is no finite trace with which to normalize the density states, and there is no operator in the algebra of the subregion that generates them*

   I suggest changing *finite trace* to *trace*. Whether the trace is finite or not is not relevant for the discussion. Note that the trace of type I$_\infty$ is not finite. I do not understand what is meant by *there is no operator in the algebra of the subregion that generates them.*

5. Page 5: *Physically, inner automorphisms are simply unitary transformations.*

   For clarity, I suggest adding *unitary transformations that belong to the algebra.* Because outer automorphisms are also unitary transformations. In fact, any automorphism of a von Neumann algebra can always be realized as a unitary transformation on the GNS Hilbert space.

6. Page 6: *it is a standard result in the theory of operator algebras that the crossed product of a type III$_1$ algebra with an outer automorphism is a von Neumann algebra of type II$_\infty$.*

   I do not think this is a correct statement. What the authors mean is the crossed product with the modular group. As a counter example, consider a theory with a finite group as global charge, e.g. $\mathbb{Z}_2$. The crossed product of the local algebra of two disjoint regions with the $\mathbb{Z}_2$ group of outer automorphisms corresponding to intertwiners is still type III$_1$.

7. Page 6: *In general, we will take it to be the Hamiltonian of the commutant $\mathfrak{A}'$.*

   I do not understand what the authors means by the Hamiltonian for the commutant algebra. Note that if $\mathfrak{A}$ is type III$_1$ then $\mathfrak{A}'$ is also type III$_1$. Note that in the case of holography, the author of [27] had a definition of this operator. But that applies only to the holographic setups. This is one of my main criticisms of this work.

8. Equation 3.8: I am not sure what this equation means. In the case of holography, there was a large $N$ parameter and we had a boundary definition of this operator. Here, the use of $G_N$ and the equation 3.13 seem ad hoc to me.

9. Section 4: As far as I can understand the authors point out the observation that Haag's duality favors entanglement wedge over causal wedge. An observation that appears and has been discussed in the literature before; for example see 2008.04810. This is often interpreted as a non-uniqueness of the choice of von Neumann algebras one can associate to GFF algebras, for example see 2210.00013. There are correspondingly two modular flows, each corresponding to one choice of von Neumann algebras. I fail to see how the crossed product construction "*represents a refinement of Haag's assignment of nets of observable algebras to spacetime regions by providing a natural construction of a type II factor*" as claimed by the authors.

The manuscript contains interesting and important discussions. However, I do not recommend it for publication in its current form.

---

## Round 2 · Referee Report · Anonymous (Referee 1) · 2024-2-14

Report

The authors have answered all my comments in a satisfactory manner.
I recommend the publication of the paper.

---

## Round 2 · Referee Report · Anonymous (Referee 2) · 2024-2-18

Strengths

The discussion is very natural and the claim of a rigor UV-finite entanglement entropy is intriguing.

Weaknesses

Currently, I have not understood why the construction leads to a regularization-independent UV-finite entanglement measure.

Report

Dear editor,

I thank the authors for the detailed response, but unfortunately, I still fail to understand equation 3.8. Defining an operator of this type is crucial for the crossed-product construction, but I do not see it can be regularization scheme independent. I am worried that the crossed-product construction ends up being regularization-dependent, in which case, we are not "paving the way to the study of entanglement entropy without UV divergences" as the authors claim.

The authors say the operator in 3.8 "... is just the standard vacuum subtraction scheme in any quantum field theory, regardless if there is a large coupling parameter or not."
Can they clarify what they mean by the "standard vacuum subtraction" for a half-boost operator?
I do not know how to "rigorously" define this operator without a regularization scheme. That is precisely why we do not have entanglement entropies in type III1 algebras of QFT. If we are willing to let go of rigor, in what sense, is this different from the conventional discussion of entanglement entropy in QFT.

It is crucial to realize that, in holography, in a sense, large N provides a natural regularization scheme. Here, we do not have that.
  • validity: good
  • significance: good
  • originality: good
  • clarity: ok
  • formatting: good
  • grammar: excellent

Author:  Ro Jefferson  on 2024-03-04  [id 4334]

(in reply to Report 2 on 2024-02-18)
Category:
answer to question

We thank the referee again for their extremely careful reading of the manuscript. We believe there are two distinct issues being conflated here: the issue of regularization (e.g., by $1/N$) and the issue of renormalization (e.g., by vacuum subtraction).

The factor of $1/N$ arises in holography because the thermofield double state belongs to the canonical ensemble. There, thermal fluctuations scale like $N^2$, hence the need to work perturbatively with $1/N$ as a regularizer. In contrast, thermal fluctuations in Rindler space die out at infinity, so there is no need for this factor (see the penultimate paragraph on page 21 and surrounding discussion). Extensive discussion of this issue can be found in 2209.10454 (ref. [34] of this version), where the authors considered a microcanonical version of the TFD. There the fluctuations are $O(1)$, so the regularization by $1/N$ is similarly not required.

The vacuum subtraction (by which we meant the standard normal ordering prescription in our previous reply) is a separate issue, and is infinite even at finite $N$ (cf. eq. (2.2) of 2112.12828 (ref. [27] in this version), where the Hamiltonian has a factor of $N$ outside a divergent integral). In analogy with a more pedagogical example, the Hamiltonian VEV here suffers from a divergence with $N$ given that it scales as $N^2$, and the usual divergence arising from the integral of the Hamiltonian density that resembles the familiar divergence for a free scalar field. This makes the vacuum subtraction used in previous approaches purely formal, since -- as the referee correctly points out -- there is no known way to rigorously define this without resorting to some regularization scheme. (We emphasize again that here we refer to some hypothetical regularization scheme for the infinite vacuum divergence, not the thermal fluctuations regularized by $1/N$ above.)

In our approach, we do *not* use this subtraction scheme, for reasons discussed in the two paragraphs immediately below eq. (3.8). We also point out the possible shortcoming of this regularization in large-$N$ gauge theory under eq. (4.6). In contrast, our choice has the advantage of avoiding the question of regularization/renormalization raised by the referee, though it is still necessary to work formally. In either case, the (state-independent) shift in the vacuum energy gets absorbed into the constant $c$, cf. the penultimate paragraph on page 11.

To more accurately reflect the need to work formally, we propose to slightly soften the claim highlighted by the referee to "*formally* paving the way to the study of entanglement entropy..." We would also like to emphasize that our construction (as other works do) relies on the fact that the crossed product of a Type III$_1$ factor by its modular automorphism group generated by the modular Hamiltonian $H_\mathrm{mod}$, which is a well-defined operator in the Type III theory unlike its half-sided counterparts, is a semifinite factor. Generally, the crossed product is a way to incorporate, on the same Hilbert space, the representation of the original algebra and the unitary action of the modular group on this algebra, neither of which makes reference to the half-sided Hamiltonians. Our central claim is true regardless of the concrete way of physically implementing this crossed product as the proposal is to replace the infinite Type III factors with their associated semifinite Type II factors obtainable via the crossed product with the modular group.

---

## Round 2 · Author Response

In this revised version, we have implemented the various clarifications suggested by the referees, and believe that this has further improved the manuscript.

---

## Round 2 · List of Changes

See the detailed reply to referee suggestions in the attached pdf.

---

## Round 3 · Referee Report · Anonymous (Referee 3) · 2024-3-15

Report

After carefully reviewing the manuscript, I concur with the assessment provided by Referee 2.
Indeed, although I think what the authors are trying to accomplish is valuable, and their line of thinking is on the right track, the idea that the cross product of a type III1 algebra with its modular flow gives a type $II_\infty$ algebra was already known in the literature, and many have speculated that this might pave the way to a rigorous notion of entropy for local algebra of quantum field theories.
Regrettably, I find that the authors have not made significant progress in advancing this idea.
Furthermore, the response provided by the authors does not adequately address the concerns raised by Referee 2.
Overall, the manuscript raises numerous unresolved questions, and while it may have merit for publication, it falls short of the standards expected for Scipost Physics.
Consequently, I recommend the publication in Scipost Physics Core instead.
  • validity: low
  • significance: ok
  • originality: low
  • clarity: good
  • formatting: perfect
  • grammar: perfect

Author:  Ro Jefferson  on 2024-03-18  [id 4374]

(in reply to Report 1 on 2024-03-15)

We thank the referee for their very prompt response. While it is true that the following statement "the crossed product of a Type III$_1$ factor with its modular automorphism group gives a semifinite, Type II$_\infty$ factor" was known in the mathematical literature and indeed was proven by Takesaki in 1973, to the best of our knowledge, we have not come across any article which relates this result to the local algebras of a generic quantum field theory, nor the regularization of their von Neumann entropies until our work which appeared in June 2023. The first paper we know which uses this result concretely is Witten's "Gravity and the Crossed Product" (which indeed was the impetus for our investigation), but this was claimed to only apply to theories with gravity and moreover did not address the issue of subregions. Indeed, generalizations, again only in the context of gravitational theories, have been made to attempt to eliminate the issue of subregions, cf. 2306.01837. In the context of the current activity in the field, the emphasis that this result is independent of gravity is important, as was recognized by referee 1, as this is clearly not appreciated by most workers in high-energy theory. In this sense, our paper does contribute something novel and important to the discussion of entanglement entropy in QFT.

We proposed the crossed product as a way to enhance Haag's net of observable algebras, basically by replacing each local Type III$_1$ with its better behaved Type II$_\infty$ factor. This accomplishes two things that are significant in the study of entanglement entropy: (1) the crossed product can serve as a regulator for generic quantum field theories and not just those with gravity, and (2) implementing this procedure for subregions is immediate in our approach. Since our result appeared, two papers have made similar claims. The first is 2306.09314, which relates the crossed product construction for a generic QFT to imposing ​"​gauge" constraints in the presence of spatial subregions, and again claims divergence resolution in entanglement entropy for general QFTs independently of gravity. The second is 2312.07646, which proposes the crossed product as a regulator for generic QFTs, once again independently of gravity, but they analyze entropy differences instead of entropies and choose an explicit embedding into the extended Hilbert space of the crossed product. Both works were posted *after* ours.

Frankly, the report we received does not go into any specifics about our work and makes general vague claims about ​"speculations" in the community that are unsubstantiated. The referee has not provided any sources for the claims that they are making regarding the novelty of our work, nor specified which "unresolved questions" they would like answered, so we are unable to reply in greater detail.

---

## Round 3 · List of Changes

Slight modification of claim in the abstract as discussed in previous exchange.

---

## Editorial Decision

published